# Reproducibility and Geometric Intrinsic Dimensionality: An Investigation on Graph Neural Network Research.

**Tobias Hille**[ORCID]*   *hille@cs.uni-kassel.de*
*Knowledge & Data Engineering Group, University of Kassel, Kassel, Germany*
*Interdisciplinary Research Center for Information System Design, University of Kassel, Kassel, Germany*

**Maximilian Stubbemann**[ORCID]   *stubbemann@ismll.de*
*Information Systems and Machine Learning Lab, University of Hildesheim, Hildesheim, Germany*
*Knowledge & Data Engineering Group, University of Kassel, Kassel, Germany*
*Interdisciplinary Research Center for Information System Design, University of Kassel, Kassel, Germany*

**Tom Hanika**[ORCID]   *tom.hanika@uni-hildesheim.de*
*Intelligent Information Systems, University of Hildesheim, Hildesheim, Germany*
*Knowledge & Data Engineering Group, University of Kassel, Kassel, Germany*
*Interdisciplinary Research Center for Information System Design, University of Kassel, Kassel, Germany*

**Reviewed on OpenReview:** *https://openreview.net/forum?id=CtEGxIqtud*

## Abstract

Difficulties in replication and reproducibility of empirical evidences in machine learning research have become a prominent topic in recent years. Ensuring that machine learning research results are sound and reliable requires reproducibility, which verifies the reliability of research findings using the same code and data. This promotes open and accessible research, robust experimental workflows, and the rapid integration of new findings. Evaluating the degree to which research publications support these different aspects of reproducibility is one goal of the present work. In order to do this, we introduce an ontology of reproducibility in machine learning and apply it to methods for graph neural networks.

The objective of this study is to try and identify hidden effects that influence model performance. To this end, we employ the aforementioned ontology to control for a broad selection of sources and turn our attention to another critical challenge in machine learning. The curse of dimensionality, which induces complication in data collection, representation, and analysis, makes it harder to find representative data and impedes the training and inference processes. The closely linked concept of geometric intrinsic dimension is employed to investigate the extent to which the machine learning models under consideration are influenced by the intrinsic dimension of the data sets on which they are trained.

**Keywords:** Reproducibility, Replication, Curse of Dimensionality, Intrinsic Dimension

## 1 Introduction

Machine learning (ML) is a rapidly evolving field that has made significant contributions to numerous industries. In view of its considerable impact, it also becomes apparent how difficult it is to replicate and reproduce empirical findings in the field of ML. Therefore, reproducibility in ML is an important topic in its own right. Reproducibility, defined as the ability of a researcher to duplicate the results of a prior study using the same materials as the original investigator, is critical to ensuring the validity and reliability of research findings. It promotes transparency, allows for verification of results, and fosters confidence in the scientific community. Despite its importance, achieving reproducibility in ML research is challenging due to several barriers. One of the main difficulties is the implementation of the exact experimental and

---

*Corresponding author

computational procedures as described in the original work. The resulting layers of complexity become particularly apparent when the used computational frameworks continually update and rise and fall in popularity and levels of maintenance. Another major challenge is the inherent instability of results. This is influenced by a multitude of factors such as the amount of data available, the computational resources at hand, the determination of hyperparameters, and the inherent randomness of the training process. In this context, it is even more difficult to assess the influence of uncontrolled epistemic uncertainties, such as the intrinsic dimensionality. Several guidelines, originating from conferences, workshops, and coding frameworks, provide recommendations and tools that help researchers and authors in this regard (Pineau, 2020; ICLR, 2019; Lightning AI and Contributors, 2022). However, these are often not very detailed or only allow a limited structural evaluation and comparability of reproducibility. Moreover, several authors note a lack of standard terminology for reproducibility (within ML), which hinders the emergence of an unified evaluation framework (Tatman et al., 2018; Bouthillier et al., 2019).

This paper proposes a comprehensive and in-depth framework for the study of reproducibility in the research area of graph neural networks. The challenges associated with *data set*, *method* and *result* are analysed in terms of their significance for computational reproducibility. A multi-stage selection process identified six scientific papers for which we studied and adapted our framework. With their help, we explore and demonstrate the limits and difficulties of reproducibility. This results in a new ontology for scientific reproducibility that generalizes to the realm of machine learning as a whole.

A second major challenge for the reproducibility of high-dimensional ML results is the occurrence of epistemic uncertainties. This is particularly the case when an attempt is made to transfer a result to new data or use cases. A particular instance of this uncertainty is the umbrella term *curse of dimensionality*. This is based on various mathematical observations in high-dimensional spaces that are generally not addressed by ML studies. A geometric approach towards understanding the *curse of dimensionality* was established by V. Pestov. He proved that the concentration of measure phenomenon (Milman, 1988; 2000) contributes to the overall *curse of dimensionality* (Pestov, 1999; 2007b;a; 2010b;a). His approach was adapted towards a practical computable function for estimating the intrinsic dimension (ID) of a geometric data set (Hanika et al., 2022). This result was further improved with regard to its applicability to large data sets (Stubbemann et al., 2023a;b).

With regard to reproducibility, we investigate the influence of the ID on the ML training process. In particular, we experiment with ID-based feature selection, as it provides a straightforward method to manipulate the ID of a data set. As we hypothesize that training methods are susceptible to ID-changes in the underlying training data set, we apply different ML methods to the same manipulated data sets. We thereby study the impact of altering the intrinsic dimension of graph data sets for all six reproduced graph neural network methods.

Although there are studies on these theoretical and practical aspects, the present work aims to bridge the gap between them by focusing on reproducibility and the intrinsic dimension within a geometric understanding. Overall, our work contributes to improving the quality and reliability of ML research, ultimately benefiting the broader scientific community and industry applications.

To summarize our contributions:

- **We introduce an ontology of reproducibility in Machine Learning** (Section 3).

- **We consider about 100 publications from the field of graph neural networks and reproduce six of them extensively** (Section 4).

- **We investigate how the change of the geometric intrinsic dimension in data sets effects the performance of the six reproduced methods** (Section 5).

## 2 Related Work

**Reproducibility and Replicability**

Several publications have investigated the general state (National Academies of Sciences, Engineering and Medicine, 2019) and challenges (Nature Special, 2018) of reproducibility and replicability in science. There is also work that has looked more specifically at these issues in the field of computer science (Freire et al., 2012) and its subfield of machine learning (Raff, 2019; Liu et al., 2020a; Chen et al., 2022). In recent years a growing number of conferences include dedicated tracks for reproducibility efforts or specific workshops (Stodden et al., 2013; ICLR, 2019). The knowledge collected there is now available in straightforward checklists (Pineau, 2020) and general reports (Pineau et al., 2021). Related to this more and more journals and publisher provide specific editorial policies (Casadevall & Fang, 2010; Springer Nature, 2020) to help authors in that regard. Efforts for reproducing and replicating past works from broad range of research fields concentrate in some dedicated journals (ReScience C, 2023), in which publications from further back are also of interest.[1] Beyond the space of academic publication there are of course similar efforts made by the programming community (paperswithcode, 2021; Sinha & Forde, 2020). Specifically, machine learning engineering teams and individuals are building frameworks (Lightning AI and Contributors, 2022) and templates (ashleve and Contributors, 2022) to streamline the process of setting up reproducible machine learning experiments. Recent research has shown that the choice of the machine learning framework used and its version (Pham et al., 2020; Shahriari et al., 2022), or the commercially available platforms providing related services (Gundersen et al., 2022b), can have a significant impact on the reproducibility properties of the research code. There are, motivated by practical concerns, surveys that investigate directly the availability and operability of research code (Collberg et al., 2015). Few works additionally try to construct a taxonomy of those reproducibility properties (Goodman et al., 2016; Kitzes et al., 2018; Tatman et al., 2018; Bouthillier et al., 2019). The accompanying discussions often emphasize the confusing terminology (Peng, 2011; Plesser, 2018; Gundersen, 2020). The provided taxonomies usually consist of a shallow hierarchy of different levels of reproducibility which are characterized by high-level features of the submissions that have to be assessed. In most cases the process of evaluating is guided by only a few questions. As such they give researchers and reviewers not that much guidance when evaluating the degree of reproducibility. However there are publications that go more into detail when analysing factors and variables that influence reproducibility (Ivie & Thain, 2018; Gundersen et al., 2018; Gundersen & Kjensmo, 2018; Gundersen et al., 2022a). This focus on central aspects of computational reproduciblity can also be found in the present work.

**Intrinsic Dimension and Feature Selection**

The term *intrinsic dimension (ID)* has multiple slightly different meanings in related sub-fields of machine learning. They share the motivating aspect of using the value of the ID of data as a proxy for gaining evidence on how the data is structured. One prominent usage of the term is for specifying the often approximated dimension of a hypothetical embedded manifold in the data space which describes almost all samples with sufficient accuracy (Hein & Audibert, 2005; Tatti et al., 2006). This notion of ID can be used to motivate a variety of estimators, for example based on sampling around data point neighborhoods (Kim et al., 2016). Those estimators give rise to different feature selection methods (Traina et al., 2010; Mo & Huang, 2012; Suryakumar et al., 2013; Golay et al., 2016), occasionally based on gradients to learn an embedding with the desired properties (Pope et al., 2021). Other work has focused on the ID of the learning process or the learned model itself. One line of research is the analysis of the ID of activations in different layers in deep neural networks (Ansuini et al., 2019; Doimo et al., 2020). There, the notion of ID is slightly adapted to be the minimum number of coordinates required to describe a set of points without significant loss of information. On the one hand, the mathematical derivations are often specifically tailored to neural networks and, on the other hand, only in a few cases do they explicitly depend on the chosen feature sets employed by the model architecture. However, these algorithms do not help to decide if and to what extent the data set is affected by the *curse of dimensionality* and the related concentration phenomena (François et al., 2007; Houle, 2013).

---

[1]See the *ten years challenge* http://rescience.github.io/ten-years/

In contrast, the *intrinsic dimension for data* (Pestov, 1999; 2007b; 2010b; 2011) gives an axiomatic approach on quantifying the direct influence of the *curse of dimensionality* by linking it fundamentally to the phenomenon of concentration of measure (Gromov & Milman, 1983; Milman, 1988; 2000). These mathematical works give rise to the intuitive view on the *curse of dimensionality* as the phenomenon of features concentrating near their means or medians, so that algorithms are therefore not able no discriminate the data. This approach by itself was computationally infeasible until the introduction of the intrinsic dimension of *geometric data sets* (Hanika et al., 2022) Although a geometric data set necessitates the inclusion of a set of feature functions, it affords the theorist or practitioner considerable flexibility in adapting the ID to emphasize a particular aspect. Consequently, the resulting ID is highly contingent upon this initial choice. For example, when placed within a geometric data set together with geodesic distances as feature functions, the Euclidean n-sphere exhibits an intrinsic dimension in the order of n. Thus for these feature functions the obtained value is consistent with other mathematical notions of dimension. The complexity of evaluating feature functions varies, and thus some classes of functions are more usable than others. Later publications provided algorithms for practical computation or approximation of this ID (Stubbemann et al., 2023a) and its application to feature selection (Stubbemann et al., 2023b), even for large-scale data sets.

**Studies Regarding Influence of Data on Model Behavior**

Machine learning models are heavily influenced by the quality and nature of the input data they are trained on. This relationship has been extensively studied in various contexts. A large body of literature is concerned with influence of *simple* data augmentation (*e.g.* cropping, rotating, stretching for computer vision data sets) when keeping a machine learning model fixed (Salamon & Bello, 2016; Perez & Wang, 2017; Tsuchiya et al., 2019; Tian et al., 2020; Laptev et al., 2020). Naturally there are also works studying influence of feature selection methods (Koçak et al., 2019) and projection methods (Wan et al., 2021) in addition to those referenced in the previous sections.

In the realm of classical machine learning, theoretical studies have been conducted on various models, including decision trees (Syrgkanis & Zampetakis, 2020) and quadratic classifiers (Latorre et al., 2021), that explore the estimation capabilities and the performance within high-dimensional settings. These models often exhibit a dependence on dimension, particularly in the context of high-dimensional regimes where their effectiveness may vary. Similarly, research has been dedicated to understanding the behavior of support vector machines in spaces with low (box-counting) dimension (Hamm & Steinwart, 2020).

In a different vein, the concept of influence function is used to track the impact of training data on learning algorithms, providing insight into how the model's predictions on test data are influenced by the training data. When applied to neural networks, influence functions shed light on the backpropagation process and the attribution of training data importance in these complex architectures (Koh & Liang, 2017; Pruthi et al., 2020; Akyürek et al., 2022; Hammoudeh & Lowd, 2022).

Otherwise, there seems to be a lack of research on data manipulation methods that focus on the influence of the concentration of measures phenomenon on model performance.

## 3    An Ontology of Reproducibility in Machine Learning

The term reproducibility is often used ambiguously and vaguely in the field of machine learning (Peng, 2011; Kitzes et al., 2018; Plesser, 2018). In our work, we apply the "classical" understanding of reproducibility in science. That is, whenever a scientific study is replicated the original experimental results should be achieved with a high degree of reliability. Of course, the concept of "replication" implies a series of attributes. Among other things, these can be the independence of the project (e.g. different group of researchers), use of different set of equipment (hardware or software), or the time since the original study was realized. As such one can see that particular scientific results can be placed on a spectrum of reproducibility. To give a more explicit process of contextualizing reproducibility we introduce in the following section an extensive *ontology of reproducibility* for the realm of machine learning. The necessity for this is based on the fact, that, to the best of our knowledge, such an ontology does not yet exist. The components of this ontology are influenced and inspired by the Chapter *Assessing Reproducibility* of the online version of the book *The*

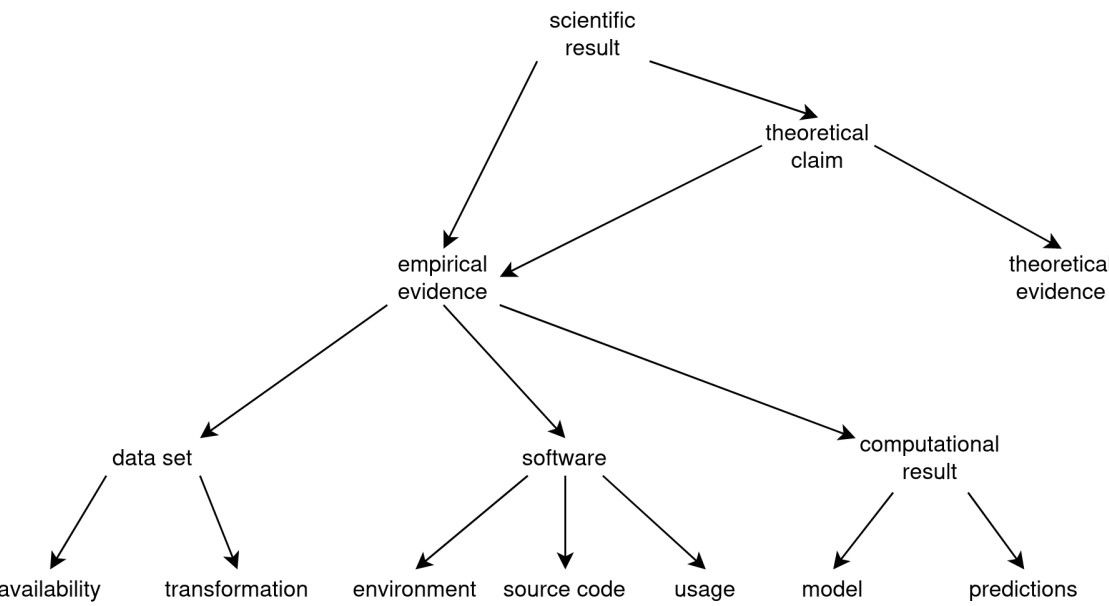

Figure 1: Top levels of the reproducibility ontology.

*Practice of Reproducible Research* (Kitzes et al., 2018). Additional ascendancy comes from existing efforts to characterize computational reproducibility (Gundersen et al., 2018; Gundersen & Kjensmo, 2018; Gundersen et al., 2022a). We adapted them with a stronger formalization, giving structure to the proposed questions and adjusted the ontology to better meet the requirements for the subsequent reproducibility study. Other noteworthy influences come from an ontology for semantic terms in machine learning (Publio et al., 2018), a practical taxonomy of machine learning (Tatman et al., 2018) which however has no formalization and very few specific points to check, and the machine learning reproducibility checklist (Pineau, 2020). One commonality between this ontology and the above-mentioned works is the focus on reproducibility of an individual research project. For simplicity, we consider only the setting where one computational result is presented as evidence for one scientific result.

## 3.1 Overview

We now present our ontology of reproducibility in machine learning. It connects possible errors or difficulties that could arise when trying to reproduce the results of a single scientific study. Our proposed ontology is structured as a hierarchy and starts on the top level with the general notion of a *scientific result.* Such a result can be based on *empirical* or *theoretical evidence.* Because we are (subsequently) interested in research that uses experiments for producing evidence we do not subdivide the theoretical category. In contrast, we propose a fine-grained structure for the empirical category. To reflect the related central aspects of data processing we use the ontological entities *data set*, *software* and *computational results* as the main subcategories for the empirical category. Each of these aggregate again a set of subentities. For example, the data set category encompasses the *availability* and *transformation* of data. Similarly, the software category has as central subcategory *source code* but also includes *environment* and *usage* as subcategories. Most importantly, we include the derived *model* and its *predictions* in the computational result category. Figure 1 shows the schema that provides an overview of the main categories and their subcategories.

We further evaluate every considered research paper within the proposed ontology based on a set of questions. In the following we want to describe the formalization of these questions and their motivations, which are based on potentially occurring errors and their impact on the scientific reproducibility. As we follow an open world semantic with our ontology the questions are formulated in such a way that answering them negatively is good for reproducibility. This also means that in the subsequent questioning, a missing answer does not indicate the non-existence of the corresponding property. We have included some questions that

remain unanswered for all the publications surveyed (a good thing), but could be helpful in obtaining a finer distinction of reproducibility when the situation they cover occurs.

## 3.2 Data Set

One central aspect of scientific result in machine learning is the data set to which previous and proposed methods are applied. A common approach is to evaluate methods over multiple different but similarly structured data sets. Reproducibility is only possible with detailed knowledge about the process for obtaining and preparing used data sets.[2] Explanations and automatization minimize the risk of working with a data set that follows a different distribution than the one used in the original study. Within this category of the ontology, we want to focus on whether the publication and accompanying material include steps (manual or automated) that describe how to obtain data sets.

### 3.2.1 Availability

The questions from this category aim to reflect nuances in obtaining and understanding the data sets used by the publication.

**D1—Is the data set format not documented?** Applicable mainly for publications that introduce a new or heavily modify an existing data set. Otherwise it is difficult to adapt methods for reproducibility.

**D2—Was the data set version not set explicitly?** Over time data sets might undergo changes. Individual samples can be relabeled, new samples included or others removed through further cleanup. These modifications lead to the need to keep track of the used version of the data set.

**D3—Was the data set not directly accessible?** Since most of the data sets currently used in machine learning are available on the web, a direct download link is a good start for reproducibility.

**D4—Did the access not work at time of study?** Unfortunately, the hyperlinks provided no longer point to the original resource sometimes.

**D5—Does the data set have privacy restrictions?** Although not very common in widely used machine learning data sets, privacy concerns may lead to access restrictions.

**D6—Does the data set require a restrictive license agreement for accessing?**

**D7—Is the data set available on request only?** This is loosely linked to the previous point. Sometimes it might be necessary to go through a more elaborate process to obtain the data set. Such steps are often quite brittle over time and are unlikely to be maintained in the long term.

### 3.2.2 Transformation

In general a data set needs to be adapted before a method can be applied. The questions from this category deal with evaluating those pre-processing steps.

**D8—Are manual steps necessary for pre-processing?** A series of manual steps to transform a data set could easily be a source of error, so an automated solution is preferred.

**D9—Is there only an incomplete description for pre-processing steps?** The provided explanations or scripts might not be enough to get the data set in the necessary form. This point is especially crucial for lesser known data sets. For such data sets, it is particularly helpful for further investigation if tools are provided for loading and accessing individual samples.

---

[2]But even then there are special cases where methods select data samples or generates them (e.g. active learning, reinforcement learning), and reproducibility of similar aspects is handicapped by other means.

**D10—Are the train, validation and test splits not clearly defined?** Usually only a part of the data set is used for training, whereas other parts are used for validation and testing. The process of allocation should be reproducible, be it through provided files or deterministic functions.

**D11—Is the number of samples not documented?** An easy way of checking one attribute of the transformation is counting the obtained samples. Additionally it gives a high-level overview over the data efficiency of the presented approach.

### 3.3 Software

The implementation and application is a central part of a proposed machine learning method. It is one aspect of the research protocol and acts as description of the executed experiments. The software code operates on one or more data sets and produces computational results. In this category we combine aspects of the code written by the authors, ancillary software, and other components crucial for reproducibility. This keeps the ontology clearly structured.

#### 3.3.1 Environment

Questions from this category deal with general behavior of the target system, which heavily influences the context of execution of the experiments.

**S1—Is the exact version of dependencies not documented?** Multiple dependencies can interact in intricate ways. This makes pinning of exact versions necessary for avoiding possible bugs connected to incompatible versions as well as prevent time consuming fixing of conflicts.

**S2—Is the specified version of dependencies not available anymore?** Depending on the age of the publication and the type of dependency used, old versions may have disappeared from the distribution channels. For smaller projects, they may no longer be maintained by the project's developers or maintainers.

**S3—Is necessary hardware unavailable?** Many specialised hardware requirements can be circumvented by simulating a computational environment with virtual machine or container images. However, this adds a time-consuming overhead when trying to achieve reproducibility, or is not feasible in a reasonable time.

**S4—Are any seeds for random number generators not set?** Multiple dependencies each can have different random number generator, where each has to be set for getting closer to reproducibility of an experiment.

**S5—Are important variables unclear?** Some settings of an experiment run (e.g. number of GPUs used) can have significant impact on results or even side effects onto other settings.

#### 3.3.2 Usage

This category of the ontology groups together aspects regarding how the experiments were started. For the set of considered machine learning papers it is not necessary to consider user input beyond starting configuration.

**S6—Is the documentation not up-to-date?** Few publications include dedicated documentation of the source code they provide. If only a simple *Readme* file is included, it should at least not be misleading for the reproducibility attempt.

**S7—Are necessary arguments not clear?** Depending on the implementation, some arguments may be required to run an experiment, but neither the defaults nor the values used are explained or provided.

**S8—Are there missing hyperparameters?** Similar to the previous question, the values of the hyperparameters for the experiments are generally important when reproducing them, as they significantly influence the result.

**S9—Are train/test scripts incomplete?** Including the individual commands of an experiment in a single file usually makes it easier to attempt reproducibility. How to start these scripts can be a source of uncertainty if certain flags or variable values used in the provided script are missing, incorrect, corrected later and/or not explained at all. In addition, if pre-processing steps are not included or other steps in the computational pipeline are missing, reproducibility is compromised. It may also be the case that not all experiments presented in the paper have scripts that run them.

**S10—Is it unclear which version of scripts was used?** It is only natural for the main source code of the release to be changed before (and after) publication to accommodate for bugs and reviews. Problems arise when these changes are not reflected in the accompanying scripts or instructions, or when multiple scripts exist for the same experiment.

### 3.3.3 Source Code

The questions in this category are concerned with the source code files that implement the ideas of the scientific experiment. We do not have separate questions regarding the availability of source code itself similar to the data set category because we only focus on papers that provide source code with a non-restrictive license.

**S11—Is there a bug that was never fixed?** Over time, authors and external contributors may find differences or errors between the original publication, its revisions and the implementation.

**S12—Are there issue solutions that were not applied?** Usually, the discovery of implementation problems is accompanied by public discussions on the website hosting the implementation. However, it may happen that the solution discussed has not been implemented, or has not been merged from an external source fork.

**S13—Was a bug fix distributed through other channels?** On the other hand, public discussions may indicate that the fix was distributed (manually) through some other channel, such as emails or direct messages to a selected/active group of participants. In these cases, it is not clear what the detailed changes are, where they are, and how they affect reproducibility.

**S14—Did the API change?** This question relates to attribute S10. We are now considering the converse scenario, wherein the entry points to core parts of the implementations have undergone alteration, yet this is not reflected in other pertinent sources, such as documentation or supplementary materials. Furthermore, the reproducibility of results is hindered when the traceability of versions is constrained by a complex and unclear history within the version control system.

**S15—Did an out of memory error occur?** As a special type of defect, it is only recorded when there are no or incorrect requirements. As our aim is to assess reproducibility in general, we do not determine the specifics that caused this error.

**S16—Are steps for one experiment missing?** If the necessary source code for an experiment is not included, the reproducibility of this experiment is more difficult to achieve.

**S17—Are parts of the source code not available?** In addition to the previous point, we want to evaluate the possible situation where the publication uses libraries or code that is not included in the source code provided. This makes reproducibility almost impossible.

**S18—Is the hyperparameter search not included?** As hyperparameter search is an integral part of experiments, it is important for reproducibility to have an explanation or process in the implementation of how the search was performed.

**S19—Is only the general idea (and no experiments) implemented?** Another reason for the lack of experiments could be that the publication simply proposes a new machine learning algorithm or a building block for an existing one.

### 3.4 Computational Result

The result obtained from a computational experiment is the evidence in support of a scientific claim. The full reproduction of a scientific result and its evaluation depends on the successful completion of the reproducibility steps for data set and software. If this is not possible, no model or predictions can be obtained for further evaluation steps. Conversely, if the supplementary material of the publication in question contains the learned model, it is still possible to perform the subsequent reproducibility steps. However, this is a rare case. Ideally, current scientific results in machine learning should be reproducible in terms of data set and software, as well as providing the learned model. This will allow for a more in-depth comparison and analysis between the reproduced and the provided models.

#### 3.4.1 Model

As outlined above, easy access to model weights is necessary for full reproducibility, especially when other factors increase the difficulty of obtaining an optimized model independently. Furthermore, the practical problem of making the model available still has no ready-made solution. There are a few existing platforms that allow the combined hosting of source code, data sets and models. Limitations can quickly become apparent when data sets and models are large, or when multiple of them are used or shared. For now, there is only one question in this category.

**R1—Are there no parameters (weights) of the obtained model provided?** As much of today's machine learning approaches use larger data sets and models, it takes more and more time and computational resources to run the proposed method. Making model weights accessible can therefore act as a kind of shortcut if the focus is on comparison with other approaches. Depending on the programming language and language and format, it gives other researchers an straightforward way to check model specifications. In addition, when reproducibility fails, it provides a way to find the cause and, more importantly, to at least verify the author's claims.

#### 3.4.2 Predictions

This category is intended to capture aspects of the outputs of the model when evaluating over the train or test data, and how these affect reproducibility. Depending on the results presented, a comparison of different evaluation metrics may be helpful, especially if the inference requires more time or resources than are available for the specific reproducibility attempt. We note that all but the last question are inherently hard to quantify.

**R2—Are there small deviation to obtained model?** We will measure the differences by comparing the central evaluation metrics between values reported by original authors with evaluated metrics on the reproduced model. We answer the question positively, when the relative difference is in the range of $\pm$1-2%.

**R3—Are strong differences in few experiments observed?** Similarly to the previous question we assign this attribute when the difference of evaluation metrics is in the range of $\pm$2-20%.

**R4—Are strong differences in almost all experiments observed?** As an extension of the previous question, we assign this attribute when almost no reasonable reasonable reproducibility of the results.

**R5—Are the claimed results only supported by small sample size?** Individual runs of a machine learning algorithm are rarely exactly reproducible, even with the best efforts to achieve reproducibility, both by the original authors and those who reproduce the work. Therefore, averaging over several runs with the same configuration of an experiment will strengthen the reliability of the result. We mark this attribute if there are less than five such similar runs per result.

**R6—Are there no predictions (outputs of classes or decisions) on the data sets?** If the original publication provides the class predictions or decisions made by the model, the reproducibility attempt provides the opportunity to examine more comprehensive metrics for differences in model behavior.

Although it is not always feasible to provide the full set of predictions (e.g. for large data sets or methods from other fields such as reinforcement learning), it could be done for non-trivial parts of the data set.

### 3.5 Limitations and Extensions

It is apparent that this ontology is designed using a basic formalization language. All connections between the entities can be read as *part of.* The authors are well aware of the *ML-Schema* (Publio et al., 2018). However, we decided not to include or build on it, because: (i) several aspects of reproducibility could not be expressed using the ML-Schema; (ii) the focus of the ML-Schema is on sharing information about machine learning algorithms, and not on the reproducibility of a scientific result. An example of this is the lack of consideration of the influence of the seed for the random number generator.

There are a number of possible modifications or extensions that we have not included in the present version of the ontology. For example, our ontology does not take into account detailed information about theoretical evidence. This is mainly motivated by the survey in Section 4, which focuses on empirical evidence. In the category of empirical evidence, the name *software* is somewhat misleading, as it also includes aspects related to hardware. One could rename this category or add *hardware* as a separate sub-category of empirical evidence. Similarly, the entity *source code* could be extended to include details of different modes of availability and documentation. A special attribute might be the use of version control software.

Certain aspects of reproducibility are not yet considered, e.g. that plots, figures and tables can be generated automatically. Authors often fail to provide the source code for the visualization. Furthermore, our ontology does not capture aspects of data provenance.

In addition, it seems that the longer the time since publication, the lower the achievable level of reproducibility. This may be due to the ageing of the hardware and software used in the experiment, which may no longer be available.

Finally, in our ontology we treat the presence or absence of an attribute categorically. Therefore, in certain cases, evaluating a research paper using our ontology is a difficult task. Conversely, any subsequent ontological operation and explanation is independent of any interpretation of numerical values.

Furthermore, it is notable that a reproducibility score for a publication, which could be derived from the evaluated attribute, is absent. By maintaining the evaluation through the ontology in a qualitative manner thus far, an ordinal measuring structure has been obtained that allows for formal comparison and ordering. Additionally, the ordinal evaluation framework allows for the explanation of differences in reproducibility or lack thereof. Nevertheless, it would be advantageous to have a scoring function in place. However, at this moment, we are unable to define a scoring function that we can guarantee will be meaningful.

## 4 Reproducibility of Major Graph Neural Network Research Results

The first main goal of the present work is to achieve a scientific overview over the state of reproducibility in the research field of graph neural networks. For this we first depict our method of candidate selection in Subsection 4.1 and thereafter discuss all our findings with respect to our reproducibility ontology.

### 4.1 Candidate Selection

The main criterion for selecting a paper was its impact on the research field of graph neural networks. In the following we describe in detail our procedural steps. As a lower bound for the publication year we selected 2016, the year of the publication of the seminal *GCN* paper (Kipf & Welling, 2016). On the other hand we considered works that were published before 2023. Most importantly we required that the paper in question has an experimental evaluation, due to the overall objective of the study to investigate the influence of intrinsic dimension. We also included research works that were only in the preprint stage.

As the citation count is an often used proxy for measuring scientific impact, and at the same time readily available, we employ it in our selection process. In detail, we use the *Semantic Scholar* (Allen Institute for

Artificial Intelligence, 2022) search engine for selecting papers based on their average citation count since their publication.[3] In our selection process we discarded all papers without publicly available source code for their experiments. We further discarded papers that either covered implementation details of software libraries or focused on applications of existing methods. Finally we refined our selection with the help of several domain experts that pointed us to important research works from the domain of graph neural networks.

Our selection process started out with 9223 unique candidates. We then calculated each paper's score and selected the 100 papers with the highest scores. We manually ignored works that did not propose a new method in the field of machine learning. This included surveys, coding frameworks, and works that only applied Graph Convolutional Network (GCN) methods to other field of science. Additionally, publications applying methods to very specific data sets and those with time-dependent or spatial data were not included.

Out of those remaining 55 results we applied the source code criterion and arrived at 42 papers. In Figure 2 we depict the yearly distribution of the number of papers (a) and their score distribution (b).

Now the following limitation of the selection method becomes more apparent. As citations are distributed over publications and there is an increasing number of papers published each year, older papers have advantage over newer ones. Conversely, the evaluation function dampens the influence of older publications to a much lesser extent.

On the other hand we wanted our selection method to reflect a "normal" search behavior of a researcher. The power-law distribution of citations is a well known property of citation networks (Price, 1965) and also somewhat expected because they are social networks which accompany the scientific process. More specifically methods of more frequently cited papers are chosen more often than those with less citations (Hazoglu et al., 2017). The power-law distribution of the citations (and subsequently the score) motivated selecting only a few papers as those publications had the majority of the impact on the field measured by the above method.

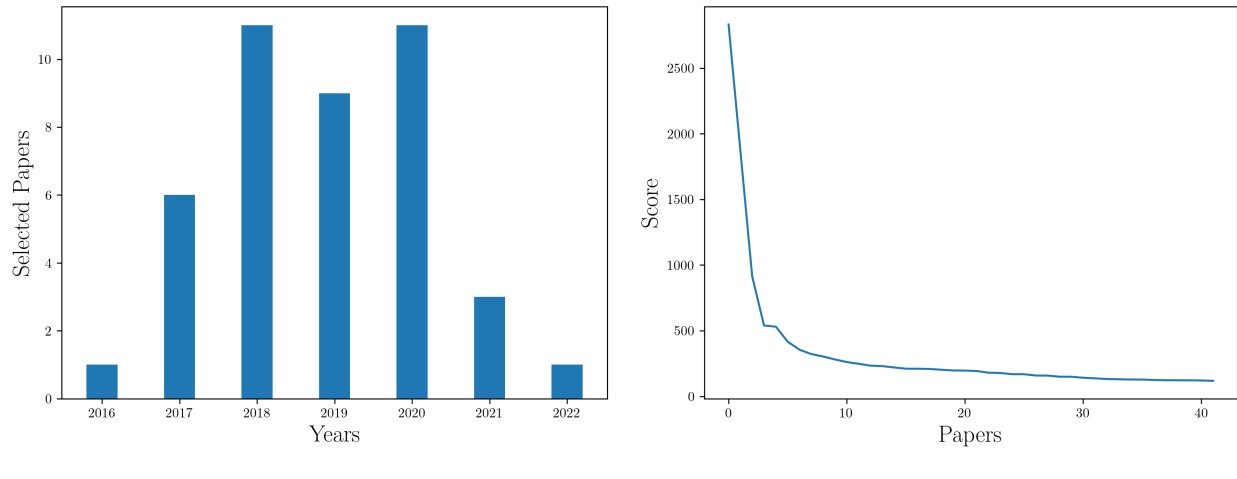

(a) Number of selected papers per year.

(b) Selected papers sorted by score.

Figure 2: Visualization to get an overview of the distribution of the selected papers.

We provide a list of considered papers in Appendix A in the appendix. We started with selecting candidates for the reproducibility survey before deciding on the automated process presented above.

The primary factors considered in assessing the reproducibility of a given paper at this stage were incompatibility of hardware requirements and errors in the source code provided. If these issues could not be resolved with reasonable effort, even with experience of using the libraries, the paper was deemed not reproducible. Unfortunately, we have not been able to fully convert the previous manual pre-selection into an process that

---

[3]In other words, we rank search results for the keyword *graph neural network* based on the score: $\frac{\text{number of citations}}{2023 - \text{year of publication}}$.

is itself reproducible. The difficulty in reproducing the results was due to the uncontrollable non-determinism introduced by the usage of the *Semantic Scholar API* over multiple runs. Furthermore new citation data changes in the rankings over time.

Making things worse, the obtained collection did not always include all papers that we tried to reproduce. However, once we got a set of considered papers, we refrained from further optimizing the selection process to include all reproduced or not reproduced papers. We acknowledge the possibility of producing a positivity bias by excluding publications where reproducibility failed completely. However, we believe this is preferable to conveying a skewed perspective of the reproducibility of the survey itself and the field of graph neural networks in general.

Therefore, the collection contains 6 publications, for which we completed the complete assignment of the described reproducibility attributes. *SGC* and *GraphSAGE* were successfully reproduced candidates determined from prior iterations of the selection process. Since we already had experience with the publication for *SAGN+SLE*, we included it as well. The final selection of reproduced papers can be seen in Subsection 4.1 together with their abbreviations used in the following.

Table 1: Reproduced Papers and the abbreviations used.

| Abbreviation | Title | Reference Key |
|---|---|---|
| GCN | Semi-Supervised Classification with Graph Convolutional Networks | Kipf & Welling (2016) |
| R-GCN | Modeling relational data with graph convolutional networks | Schlichtkrull et al. (2018) |
| GraphSAGE | Inductive Representation Learning on Large Graphs | Hamilton et al. (2017) |
| DiffPool | Hierarchical Graph Representation Learning with Differentiable Pooling | Ying et al. (2018) |
| SGC | Simplifying Graph Convolutional Networks | Wu et al. (2018a) |
| SAGN+SLE | Scalable and Adaptive Graph Neural Networks with Self-Label-Enhanced training | Sun & Wu (2021) |

## 4.2 General Observations with Respect to our Ontology

Reproducing experimental results from the selected scientific research was challenging due to various factors. It is usually the case that the papers alone do not provide enough information to replicate the experiments independently. Accordingly, we have always started the reproducibility attempt with the associated source code. One major issue is that those repositories often lack crucial dependency information (S1), making it difficult to even run the entry point scripts without errors. To overcome these challenges, it was necessary to search through the accompanying discussions and seek clarification on the exact parameters and commands that might be missing or not working correctly. Additionally, it is often the case that the commands provided in the simple documentations for running experiments rarely work as expected (S6, S7). Furthermore, the availability of different data sets adds complexity to the reproducibility process, especially considering that many publications were published before coordinated efforts to unify the data set landscape, for example the *Open Graph Benchmark* (Hu et al., 2020a). However, data sets are generally accessible, although it is rarely the case that the preprocessing steps are explained (D9). Another common point is the aspect of hyperparameter search, which is not included in most of the software provided, even if it is mentioned in the publication (S18). Finally, it is common for papers to fail to provide the model (R1) or the predictions of the model on specific data (R6). The following is a more detailed description of specific problems, grouped under the main categories of *data set, software* and *computational result*, encountered during the reproducibility attempts. For a better overview, we will focus on selected points that stand out.

We would like to emphasize that in our ontology it is *not* advantageous to have an attribute, as this is evidence that it is more difficult to reproduce. In cases where it was not clear whether an attribute was present, we chose not to disclose it.

### 4.2.1 Category: Data Set

GCN:   We observe that the data set used in the research is conveniently available as it is included in the repository. However, there is a lack of explanation regarding the preprocessing steps (D9). Despite this, a working function is provided, which can be used for the transformation process.

R-GCN:   A similar point as in GCN regarding (D9) applies.

GraphSAGE:   The availability of the *web of science* data set is limited to those with the corresponding license (D6) and upon request (D7). Additionally, manual preparation (download) of the data sets is required (D8).

DiffPool:   The implementation does not use proper train/test splits because the approach is only evaluated with k-fold validation (D10).

SGC:   A similar point as in GCN regarding (D9) applies.

SAGN+SLE:   Except of the general observations of missing explanation of the preprocessing steps (D9) the aspects of the data set category are sufficiently reproducible.

Table 2: Observations for *data set* category with respect to 4.2.1.

| | data set | | | | | | | | | | |
|---|---|---|---|---|---|---|---|---|---|---|---|
| | availability | | | | | | | transformation | | | |
| | metadata | | download | | | | | preprocessing | | selection | |
| | D1 | D2 | D3 | D4 | D5 | D6 | D7 | D8 | D9 | D10 | D11 |
| GCN | | | | | | | | | × | | |
| Relational GCN | | | | | | | | | × | | |
| GraphSage | | | | | | × | × | × | × | | |
| Diffpool | | | | | | | | | | × | |
| SGC | | | | | | | | | | | |
| SAGN+SLE | | | | | | | | | × | | |

### 4.2.2 Category: Software

GCN:   The dependencies are not properly specified (S1), which made it challenging to set up and run the experiments. It is worth noting that the documentation is not up to date (S6) and contains misleading information.

R-GCN:   Firstly, there is no requirements file provided (S1), making it challenging to recreate the necessary environment. Additionally, information about the specific Python interpreter version used is hidden. Furthermore, the seeds for randomization are not set (S4).

GraphSAGE:   The necessary arguments for the evaluation scripts are not stated (S7), leaving researchers unsure of the required inputs. Furthermore, there are discussions about possible values, adding ambiguity to the code (S8). Additionally, the evaluation scripts themselves are incomplete or misleading, further hindering reproducibility (S9). The software used in the study has some bugs that affect reproducibility (S11). For example, the evaluation script for the *ppi* data set is incomplete, but a fix is available in pull requests (S12).

DiffPool:   Unfortunately there is no explicit list of necessary requirements (S1) and only a minimal README file (S6). Additionally it seems that the provided commands do not work (S7, S8 and S9) and that the seeds for randomization are not set before the experiments (S4). The implementation also did not include steps to reproduce two experiments with the *reddit-12k* or *collab* data sets (S16).

SGC:   No features of that category that hindered the reproducibility were observed.

SAGN+SLE:   Again, there is no requirements file provided (S1). Unfortunately we encountered out-of-memory errors (S15) when trying to reproduce experiments using data sets *ogb-papers* and *ogb-mag*. It could

be argued that this would mean that the necessary hardware is unavailable (S3) but maybe it could be fixed by changing hyperparameters like batch size.

Table 3: Observations for *software* category with respect to 4.2.2.

| | software | | | | | | | | | | | | | | | | | | |
|---|---|---|---|---|---|---|---|---|---|---|---|---|---|---|---|---|---|---|---|
| | environment | | | | | usage | | | | | source code | | | | | | | | |
| | dependencies | | | variables | | documentation | | | scripts | | bugs | | | | | experiments | | | |
| | S1 | S2 | S3 | S4 | S5 | S6 | S7 | S8 | S9 | S10 | S11 | S12 | S13 | S14 | S15 | S16 | S17 | S18 | S19 |
| GCN | × | | | | | × | | | | | | | | | | | | × | |
| Relational GCN | × | | | × | | | | | | | | | | | | | | × | |
| GraphSage | | | | | | | × | × | × | | × | × | | | | | | × | |
| Diffpool | × | | | × | | × | × | × | × | | | | | | | | × | × | |
| SGC | | | | | | | | | | | | | | | | | | | |
| SAGN+SLE | × | | | | | | | | | | | | | | × | | | × | |

### 4.2.3   Category: Computational Result

GCN:   When examining the results, it is observed that there are small deviations in the test set accuracy, typically within a range of $\pm 1\%$ (R2). However, it appears that only a single run was conducted for each experiment, as no standard deviation is provided for the final performance. Although the authors claim to have run the experiments with multiple seeds, there is no evidence of this in the code, which raises concerns about the robustness of the reported results (R5). We were not able to reproduce the experiments with the *neil* data set because of the difficulties to prepare and use the data set.

R-GCN:   A similar observation regarding multiple runs was made for this publication as well. Additionally, the results of the study exhibit both small deviations and strong differences in different parts. For the AIFB, MUTAG, and AM data sets, small deviations of approximately $\pm 2\%$ in accuracy are observed (R2). However, for the BGS data set, a significant difference of 15% is observed, indicating a substantial variation in the results (R4).

GraphSAGE:   The results of the study exhibit small deviations, typically within a range of $\pm 2\%$, for the available data sets (R2). However, the code indicates that the experiments were only run once (R5).

DiffPool:   Due to time constraints we were only able to reproduce the experiments for the DD and Enzymes data set and observed small deviations $\sim 2\%$ (R2). Even though k-fold validation was used the experiment was only run once (R5).

SGC:   No feature other than the usual were observed.

SAGN+SLE:   Similar to other survey candidates we obtain results that exhibit small deviations, typically within a range of $\pm 2\%$, for the available data sets (R2).

Table 4: Observations for *computational result* category with respect to 4.2.3.

| | computational result | | | | | |
|---|---|---|---|---|---|---|
| | model | predictions | | | | |
| | | | | | | |
| | R1 | R2 | R3 | R4 | R5 | R6 |
| GCN | × | × | | | × | × |
| Relational GCN | × | × | × | | × | × |
| GraphSage | × | × | | | × | × |
| Diffpool | × | | | | × | × |
| SGC | × | | | | | × |
| SAGN+SLE | × | × | | | | × |

### 4.3 Discussion

Regarding the reproducibility ontology, we observed that most paper look almost the same in the *data set* category, but required quite different efforts in the reproducibility attempt. This is particularly visible in the fact that attributes D1 to D5 are not present in any of the surved articles. The ease of reproducibility was mainly decided by the information provided in the README document of the source code. This means that the ontology does not capture this aspect very well for this category, even if there is a corresponding attribute in the source code category.

The *software* category, on the other hand, allowed for a very good differentiation of the different papers in terms of their reproducibility. Again, some attributes do not seem to contribute to the decision on the degree of reproducibility. However, further attempts to reproduce other publications may find these currently unused attributes useful.

The *computational result* category has some attributes that are common to all reproduced papers. This suggests that the corresponding properties (providing model weights and predictions) are the most difficult to obtain. We observed several papers where the number of repetition for the experiment in the original paper was low. This could be due to higher computational requirements. We also found that we had included too few cases of possible evaluation scenarios, and that those that were included were too broadly defined. Furthermore, there are no attributes to assess the degree of reproducibility of follow-up or downstream tasks, which are also addressed in the original papers.

Several of the attributes analysed reflect the technical barriers to reproducing the work. As it is rare for a single script to start all the experiments, the necessary commands have to be collected manually. Finding the right combination of Python environments and libraries is also difficult due to the lack of specified versions and dependencies. Some methods require significant computing resources and time, so that some errors are only detected after a long time. Sometimes the logs generated are only available in the console, making debugging and monitoring difficult. In addition, checkpoints are not specific to hyperparameters, leading to potential overwriting during training sessions. This lack of specificity increases the effort required to reproduce graph neural network research and compromises the reliability of experiments.

## 5 Influence of Intrinsic Dimensionality on Model Performance

The second main goal of the present work is to investigate the influence of intrinsic dimensionality on model behavior. We begin with the mathematical foundations of the concept of geometric ID in Subsection 5.1 and then present our experiments and results.

As already mentioned the geometric intrinsic dimension (Hanika et al., 2022) is a computational accessible approach for measuring how a given data set is affected by the *phenomenon of concentration of measure* (Gromov & Milman, 1983; Milman, 1988; 2000), which itself is deeply connected to the *curse of dimensionality* (Pestov, 1999; 2007b;a; 2010b;a). Of central importance are feature functions that *concentrate*. This means that they map most of the values of their domain close to the mean or median of their image set. Pestov has postulated that features of this type contribute the most to the *curse of dimensionality*. In his approach, all 1-Lipschitz functions are considered as potential feature functions. In the revised axiomatic system introduced by Hanika et al. (2022) the notion of a *dimension function* emerged. Such a dimension function allows for estimating the extent to which the provided features concentrate on the data set without the necessity of evaluating all possible feature functions. This is especially important in machine learning, where algorithms and, in particular, models, usually only have access to a limited range of features of a given type.

In recent works, the computation and approximation of the dimension function have been improved when choosing the set of all component projections as feature functions for data sets in Euclidean space (Stubbemann et al., 2023a). The objective of this study is to build upon the findings of previous research that employed the geometric intrinsic dimension for feature selection (Stubbemann et al., 2023b).

### 5.1 Foundations of the Concentration-based Intrinsic Dimension

We commence by providing a brief overview of the mathematical definitions that the intrinsic dimension builds upon. Readers who wish to gain a more detailed understanding of this topic are directed to the cited works, which offer in-depth explanations. Since this work is primarily concerned with practical (and thus finite) setting, the following definition, which is specific to that context, will suffice for our purposes.

**Definition 1** (Adapted from (Hanika et al., 2022)). Let $\mathcal{D} = (X, F, \mu)$ be a triple consisting of a finite set $X$ of *data points* and a finite set $F \subseteq \mathbb{R}^X$ of *feature functions* from $X$ to $\mathbb{R}$. Consider the function $d_F(x, y) := \sup_{f \in F} |f(x) - f(y)|$. We require that $\sup_{x,y \in X} d_F(x, y) < \infty$ is fulfilled and $(X, d_F)$ is a complete and separable pseudo metric space with $\mu$ being the normalized counting measure on $(X, d_F)$. We call $\mathcal{D}$ a *(finite) geometric data set*.

We will now introduce the building blocks that give rise to a dimension function that fulfills the aforementioned axioms postulated in Hanika et al. (2022). A function of this nature will indicate a geometric data set by a low value, precisely when the contained data points can be more effectively discriminated by the corresponding set of feature functions.

Given a feature $f \in F$ we want to evaluate how it can discriminate sets of a specific measure (e.g. size $c_\alpha := \lceil |X|(1 - \alpha) \rceil$) for a fraction $\alpha \in (0, 1)$ of the whole $X$. For this we use can use the following function:

$$\text{PartialDiameter}\,(f, 1 - \alpha)_{\mathcal{D}} = \min_{\substack{M \subseteq X \\ |M| = c_\alpha}} \max_{x,y \in M} |f(x) - f(y)|.$$

By considering all feature from the feature set $F$ we arrive at the

$$\text{ObservableDiameter}(\mathcal{D}, -\alpha) := \sup_{f \in F} \text{PartialDiameter}(f, 1 - \alpha)_{\mathcal{D}}.$$

When considering all possible values for $\alpha$ we obtain a way to describe the ability of a feature set $F$ of a geometric data set to discriminate data points in $X$:

$$\Delta(\mathcal{D}) := \int_0^1 \text{ObservableDiameter}(\mathcal{D}, -\alpha)\, \mathrm{d}\alpha$$

It turns out that we need one more step to get the *dimension function* we are looking for:

$$\partial(\mathcal{D}) := \frac{1}{\Delta(\mathcal{D})^2}$$

For the case of finite geometric data sets it follows that the ID can be explicitly calculated with the help of the following expression

$$\Delta(\mathcal{D}) = \frac{1}{|X|} \sum_{k=2}^{|X|} \max_{f \in F} \min_{\substack{M \subseteq X \\ |M| = k}} \max_{x,y \in M} |f(x) - f(y)|.$$

Using the notation $\phi_{k,f}(\mathcal{D}) := \min_{M \subseteq X, |M|=k} \max_{x,y \in M} |f(x) - f(y)|$, and $\phi_k(\mathcal{D}) := \max_{f \in F} \phi_{k,f}$, this can be rewritten as

$$\Delta(\mathcal{D}) = \frac{1}{|X|} \sum_{k=2}^{|X|} \max_{f \in F} \phi_{k,f}(\mathcal{D}) = \frac{1}{|X|} \sum_{k=2}^{|X|} \phi_k(\mathcal{D}).$$

#### 5.1.1 Approximation of Intrinsic Dimension

The straightforward computation of the equations in the previous section is hindered by the task to iterate through all subsets $M \subseteq X$ of size $k$. This yields an exponential complexity with respect to $|X|$ for computing $\Delta(\mathcal{D})$. As suggested by Hanika et al. (2022) and later proven by Stubbemann et al. (2023a), we can instead use algorithms with a quadratic runtime complexity in $|X|$ to compute the ID. Furthermore, for settings where a quadratic runtime is still not sufficient, the authors propose the following concept.

Let $s = (2 = s_1, \ldots, s_{l-1}, s_l = |X|)$ be a strictly increasing and finite sequence of natural numbers. We call $s$ a *support sequence* of $\mathcal{D}$. We additionally define

$$\Delta(\mathcal{D})_{s,-} := \frac{1}{|X|} \left( \sum_{i=1}^{l} \phi_{s_i}(\mathcal{D}) + \sum_{i=1}^{l-1} \sum_{s_i < j < s_{i+1}} \phi_{s_i}(\mathcal{D}) \right),$$

$$\Delta(\mathcal{D})_{s,+} := \frac{1}{|X|} \left( \sum_{i=1}^{l} \phi_{s_i}(\mathcal{D}) + \sum_{i=1}^{l-1} \sum_{s_i < j < s_{i+1}} \phi_{s_{i+1}}(\mathcal{D}) \right)$$

and call accordingly $\partial(\mathcal{D})_{s,-} := \frac{1}{\Delta(\mathcal{D})_{s,+}{}^2}$ the *lower intrinsic dimension* of $\mathcal{D}$ and $\partial(\mathcal{D})_{s,+} := \frac{1}{\Delta(\mathcal{D})_{s,-}{}^2}$ the *upper intrinsic dimension* of $D$.

This results in giving us lower and upper bounds for $\Delta(\mathcal{D})$ and thus for the ID. By using upper and lower bounds, we can obtain the following approximation of the ID:

$$\partial(\mathcal{D}) \simeq \partial(\mathcal{D})_s := \frac{\partial(\mathcal{D})_{s,+} + \partial(\mathcal{D})_{s,-}}{2}.$$

Stubbemann et al. (2023a) provides an algorithm for calculating this approximation.

## 5.2 Dimension based Feature Selection

The intrinsic dimension of a data set refers to a measure of concentration that captures the underlying structure or information of the data. It is challenging to quantify the impact of intrinsic dimensionality on a particular machine learning method. This motivates the need to investigate its effect. One approach to achieving this is by discarding the features that have the most significant influence on the dimensionality of the data set. The removal of these features allows for the observation of any changes in the performance of the trained model. This approach allows us to examine the relationship between intrinsic dimensionality and model performance. Feature selection can be regarded as a means to an end in this research. It serves as a tool to identify and eliminate those features that contribute the most to the dimensionality of the data set. In order to calculate the influence of dimensionality and perform feature selection, we rely on methods demonstrated in Stubbemann et al. (2023b), which we will briefly include in the following paragraphs.

The *discriminability of $\mathcal{D}$ with respect to feature $f \in F$* is defined as

$$\Delta(\mathcal{D})_f^* := \frac{1}{|X|} \sum_{k=2}^{|X|} \phi_{k,f}(\mathcal{D}).$$

Note, that one data point with an outstanding value $f(x)$ can have a strong influence on $\Delta(\mathcal{D})_f^*$ via drastically increasing $\phi_{|X|,f}(\mathcal{D})$. To weaken this phenomenon, we weight $\phi_{k,f}(\mathcal{D})$ higher for smaller values of $k$.

The *normalized discriminability of $\mathcal{D}$ with respect to $f$* which we define as

$$\Delta(\mathcal{D})_f := \frac{1}{|X|} \sum_{k=2}^{|X|} \frac{1}{k} \phi_{k,f}(\mathcal{D}).$$

The *normalized intrinsic dimensionality of $\mathcal{D}$ with respect to $f$* is then given by

$$\partial(\mathcal{D})_f := \frac{1}{\Delta(\mathcal{D})_f^2}.$$

The higher this value is for a given feature, the more it contributes to the intrinsic dimension and thus reduces the ability to discriminate between data points.

Stubbemann et al. (2023b) provides an algorithm for calculating the normalized intrinsic dimensionality directly.

### 5.2.1 Approximation of Discriminability

Unfortunately, as before, an explicit calculation of the normalized discriminability for larger data sets is not feasible because the algorithm scales quadratically with the number of data points. We can, however, use a similar approach to the previously referenced method of approximating the intrinsic dimension with the help of support sequences to approximate the discriminability as well.

For a feature $f \in F$ and a support sequence $s$ we call

$$\Delta(\mathcal{D})_{s,f}^+ := \frac{1}{|X|} \left( \sum_{i=1}^l \frac{1}{s_i} \phi_{s_i,f}(\mathcal{D}) + \sum_{i=1}^{l-1} \sum_{s_i < j < s_{i+1}} \frac{1}{j} \phi_{s_{i+1},f}(\mathcal{D}) \right)$$

the *upper normalized discriminability with respect to $f$ and $s$* and

$$\Delta(\mathcal{D})_{s,f}^- := \frac{1}{|X|} \left( \sum_{i=1}^l \frac{1}{s_i} \phi_{s_i,f}(\mathcal{D}) + \sum_{i=1}^{l-1} \sum_{s_i < j < s_{i+1}} \frac{1}{j} \phi_{s_i,f}(\mathcal{D}) \right)$$

the *lower normalized discriminability with respect to $f$ and $s$.*

We define the *upper/lower normalized intrinsic dimensionality with respect to $f$ and $s$* via $\partial(\mathcal{D})_{s,f}^+ := \frac{1}{\left(\Delta(\mathcal{D})_{s,g}^-\right)^2}$ and $\partial(\mathcal{D})_{s,f}^- := \frac{1}{\left(\Delta(\mathcal{D})_{s,f}^+\right)^2}$. Equipped with these we then can assign each feature their *approximated normalized intrinsic dimensionality with respect to $f$ and $s$*:

$$\partial(\mathcal{D})_f \simeq \partial(\mathcal{D})_{s,f} := \frac{\partial(\mathcal{D})_{s,f}^+ + \partial(\mathcal{D})_{s,f}^-}{2}.$$

Stubbemann et al. (2023b) provides an algorithm for calculating this approximation of the normalized intrinsic dimensionality.

### 5.3 Experimental Execution and Impact on Intrinsic Dimension

In order to demonstrate the impact of the intrinsic dimensionality of the different data sets on the methods employed in the reproduced papers, we have chosen to discard those features with the highest (approximate) normalized intrinsic dimensionality.

In the context of contemporary machine learning, data sets are typically represented by matrices. For graph data, this typically refers to the data pertaining to the nodes $X$. In addition, the connectivity information, represented by the adjacency matrix $A$, and any edge features, are also considered. However, aggregating this information into a feature matrix through the use of neighbourhood aggregation, in the form of $A^k X$ (where $k$ is a small positive integer), does not alter the qualitative insights provided by the intrinsic dimension, as demonstrated by previous research (Stubbemann et al., 2023b). Aggregation is a common feature of graph neural network methods, which often employ indirect forms of aggregation through the use of specific models. Given this, we have chosen to refrain from considering the neighbourhoods, and instead focus on the matrix of node features of shape $n \times d$, where $n$ indicates the number of samples and $d$ the number of attributes per sample. For each data set in our investigation we use the following representation as a geometric data set as introduced in Definition 1. The set $X$ is comprised of the $n$ samples $x_i$ where each sample consists of the attributes $x_i = (x_{i1}, \ldots, x_{id})$. We chose the set of component selectors $f_j(x) = x_j$ as the set of feature functions $F$. Together with the counting measure $\nu(A) = |A|/n$ for a subset $A \subseteq X$ we complete our special instance of the geometric data set $\mathcal{D}$.

The sizes of all used node feature matrices can be seen in Table 5.

Table 5: Sizes for all data sets and the research works they appear in, in the scope of this work.

| Data Set Name | Nodes | Edges | Features | Paper Names |
|---|---|---|---|---|
| citeseer | 3312 | 4732 | 3703 | GCN, SGC |
| cora | 2708 | 5429 | 1433 | GCN, SGC, SAGN+SLE |
| pubmed | 19717 | 44338 | 500 | GCN, SGC |
| aifb | 8285 | 29043 | 4 | R-GCN |
| mutag | 23644 | 74227 | 2 | R-GCN |
| bgs | 333845 | 916199 | 2 | R-GCN |
| am | 1666764 | 5988321 | 11 | R-GCN |
| enzymes | 19474 | 37282 | 18 | DiffPool |
| ppi | 14755 | 225270 | 50 | SAGN+SLE |
| ppi (large) | 56944 | 818716 | 50 | GraphSAGE, SAGN+SLE |
| reddit | 232965 | 11606919 | 602 | GraphSAGE, SGC, SAGN+SLE |
| flickr | 89250 | 899756 | 500 | SAGN+SLE |
| yelp | 716847 | 6977410 | 300 | SAGN+SLE |

**Data set preparation**  For each research paper, we initially extract the essential components for loading and preprocessing the data sets from the source code supplied by the authors. These components are then used to obtain the node feature matrices of the data sets used. In instances where it is unavoidable, we resort to concatenating the node feature matrices from both the training and test data. It is of paramount importance to note that we rigorously ensure that the machine learning method does not have greater access to the test data than was originally permitted in the original implementation.

The rationale for applying feature selection after preprocessing is as follows: Our approach aims to investigate how methods are influenced by the data on which they are applied, for example, how the model "sees" the data. Some forms of preprocessing alter the empirical data distribution, and preprocessing typically lacks any learnable parameters. Furthermore, the model in question rarely has explicit information about the applied preprocessing steps. Consequently, we do not consider the preprocessing steps to be part of the model. This approach facilitates the separation of the influence, as otherwise the change in the preprocessing resulting from feature selection would be included in the observations and discussions.

**Feature selection**  We used the algorithm for direct calculation of the discriminability (Stubbemann et al. (2023b), Algorithm 1) for data sets with less than $10^5$ samples. For larger data sets, we employed the approximating version (Stubbemann et al. (2023b), Algorithm 2). In those cases we first choose a geometric sequence $\hat{s} = (s_1, \ldots s_l)$ of length $l = 10,000$ with $s_1 = |X|$ and $s_l = 2$ and use the support sequence (Subsection 5.2.1) $s$ which results from $s' = (\lfloor |X| + 2 - s_1 \rfloor, \ldots, \lfloor |X| + 2 - s_l \rfloor)$ via discarding duplicated elements. We then discarded for every factor $\alpha \in \{0.1, 0.2, \ldots, 0.9\}$ the corresponding fraction of the features with highest (approximated) normalized intrinsic dimensionality from all data points. Following the selection process, the machine learning algorithms from the relevant papers are applied to the feature-reduced data sets using the same (hyper-)parameter configuration as the original. In order to facilitate an objective evaluation, the same scores (accuracy or f1) as those originally produced were collected over repeated training runs with ten different seeds.[4] We did not test other feature selection methods as similar investigations were already done in Stubbemann et al. (2023b).

### 5.4  Observations

We present in this section the computational results and observations for the experiment. Here we focus on the details corresponding to the two research works GCN and SAGN+SLE. Afterwards we will state general observations for the remaining experiments, but refer the reader to Appendix C for accompanying plots.

In Figure 3a we show the analysis for the intrinsic dimensionality of the three data sets from GCN. Because the data sets are differently sized we need to find a common representation. First we order the feature set for each data set using the normalized intrinsic dimensionality of each feature as the score. On the x-axis we

---

[4]An exception was the diffpool enzymes experiment, where only a smaller number of runs was feasible given the runtime of the algorithm.

give the relative position of the sorted feature set, i.e., position $\alpha$ indicates that the corresponding feature is at the sorted position $\alpha \cdot |F|$. As the measured normalized intrinsic dimensionality can vary widely between the data sets we decided to normalize it by dividing, for each data set $\mathcal{D}$, the value $\partial(\mathcal{D})_f$ by $\max_f \partial(\mathcal{D})_f$. The corresponding values are depicted in the y-axis in Figure 3a.

We observe that all curves increase monotonically in value with respect to the ranked position of the features. This is expected as we sort by this value. However, the slope is solely dependent on the individual contributions of the features to the intrinsic dimension. The stair case pattern is not an artifact of the plot but rather results directly from the data set and its preprocessing. This indicates that a lot of features have the same normalized intrinsic dimensionality per step.

We further observe that the *pubmed* data set (green) entails features with a similar high normalized intrinsic dimension. Or more general, the higher the line in the plot the more similar are the values of the individual features of a data set $\mathcal{D}$ compared to the maximal feature value $\max_f \partial(\mathcal{D})_f$. This allows for comparing the feature behavior of the different data sets. For example, with respect to this property we observe that the *cora* data set (orange) has more diverse distributed features compared to the *pubmed* or *citeseer* data set (blue).

Figure 4a demonstrates that the distributions for the normalized intrinsic dimension of the data sets used for the SAGN+SLE method are of greater variety than those discussed earlier. We can see that the *cora* data set does not have a prominent stair case pattern, which can be explained by different preprocessing steps that smooth the features relative to each other. One standout distribution is that of the *reddit* data set, which is shaped like a hockey stick.

As we calculated the (approximated) normalized intrinsic dimensionality on the data sets, occasionally different normalized rankings for what seems to be the same data set emerged through different steps of their preprocessing. A highly visible example can be found in Appendix C with the *reddit* data set in the GraphSAGE (Figure 8a) and SGC (Figure 9a) experiments.

**Accuracy and Intrinsic Dimension**  Figure 3b shows the accuracy of the resulting model when applying the GCN machine learning method to the feature reduced data sets. In this experiment, both the training and test data sets are feature-reduced. This ensures that the algorithm is trained and tested on the same set of features.

For each data set we reduced the number of features in steps of 1% up until 10% was reached. Here the percentage steps are taken with respect to the size of the complete feature set $F$. Afterwards we continued with 10% steps. Both are indicated on the x-axis in Figure 3b. After training the corresponding model we collected the obtained accuracy, similar to the original work. To achieve a meaningful estimate for the model behavior with regard to our dimensional data set perturbation, we measure the average over ten identical runs with different seeds. The resulting standard deviation is shown via error bars in the plot. Similar plots for the other papers can be found in Appendix C.

We observed that the methods sometimes failed to converge for smaller discarding values ($< 0.01$). This behavior was very irregular and we did not include these runs and their corresponding discarding values in the figures. Our investigation into the causes showed that this was usually due to some artifacts of the machine learning method, such as early stopping.

Additionally we conduct further experiments with random or reversed feature selection, where the latter means the discarding of features with the lowest normalized intrinsic dimensionality first. For the sake of completeness, by random we refer to the process of randomly selecting features from $F$. For all discarded data sets we applied the GCN method with ten different seeds. Due to the expected long run times, these extended experiments (random/reversed, 1% discard steps) were not conducted for methods other than GCN.

For all data sets the resulting model performances are relatively stable under the aforementioned primary discarding method. Yet when applying the reverse selection method a fast deteriorating performance can be observed. This behavior starts already at the smallest discarded proportion and is very pronounced.

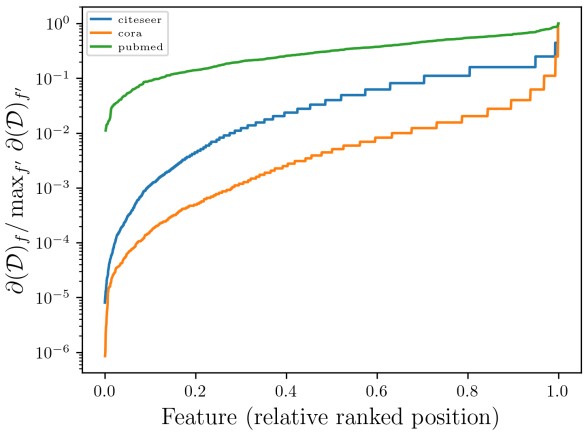

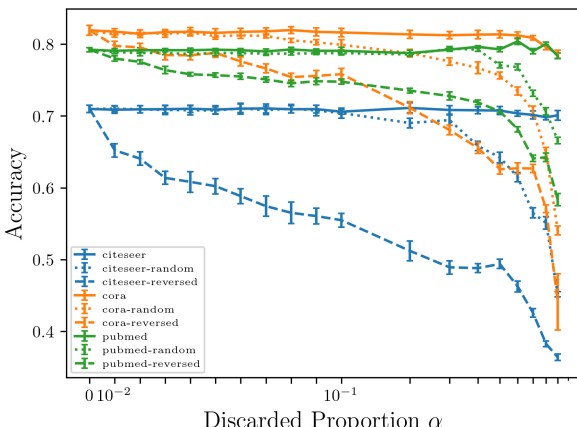

(a) Normalized intrinsic dimensionality (y-axis) of the *cora*, *citeseer*, and *pubmed* data sets plotted against relative ranked position of features (x-axis). For every data set $\mathcal{D}$ the sorting key is defined by normalized intrinsic dimensionality divided by $\max_f \partial(\mathcal{D})_f$. The values themselves are normalized by the highest value and sorted in ascending order.

(b) Accuracy of the resulting model (y-axis) after altering the GCN data sets based on feature selection. We discarded a fraction $\alpha$ of features (x-axis) with the highest normalized intrinsic dimensionality from the original data set. Curves labeled with *random* or *reversed* used a random or reversed selection method respectively. Bars indicate standard deviation over ten repetitions with different seeds.

Figure 3: Influence of Intrinsic Dimension measured through feature selection for the GCN results.

A more intricate detail can be observed for the random discarding method by combining the information from the two Figures 3a and 3b. We find that the higher the line in Figure 3a the later (i.e., higher values of $\alpha$) the break off between performance of normal and random discarding in Figure 3b.

We also see small fluctuations and drops in performance at the highest discarding values. This becomes more apparent when directly visualizing the differences to the proposed discarding method.

This picture changes slightly when looking at the results related to the SAGN+SLE experiments in Figure 4b. For some data sets the same stagnating behavior is evident. For others, however, there is a marked drop in performance. Especially for the two *ppi* data sets there is a greater variation in performance. Another different behavior can be seen for the *yelp* data set, where the performance starts to decrease for lower discard factors.

### 5.5 Discussion

We now want to contextualize the observations and results. The Figures 3a and 4a show the distributions of normalized intrinsic dimensionality (NID) and we observe that distinct values arise for different data sets. We may note that the figures show relative and not absolute NID and therefore their respective values should not be compared. Furthermore, even in the absence of this relative scaling, the mathematical modelling of the intrinsic dimension permits a direct comparison.

For our discussion we compare the different values of NID to the performance of the corresponding models, as shown in Figures 3b and 4b. From this we can infer the following link. consider the difference between the lowest and the highest value of the NID for a given data set. We find that this difference decreases in the following order: *pubmed, citeseer, cora*. If we look to the corresponding model performance in Figure 3b, we observe that the performance of the random feature selection divergences for different proportions $\alpha$. Interestingly this happens in the same order as before, albeit at different levels of accuracy. The difference between the lowest and highest values of NID indicates that the individual contributions to the ID by the corresponding features is more evenly distributed. For example, in the case of a horizontal line every feature contributes equally to the ID. Conversely, a ⌐⌐-shape, as observed in Figure 4a for the *reddit* data set,

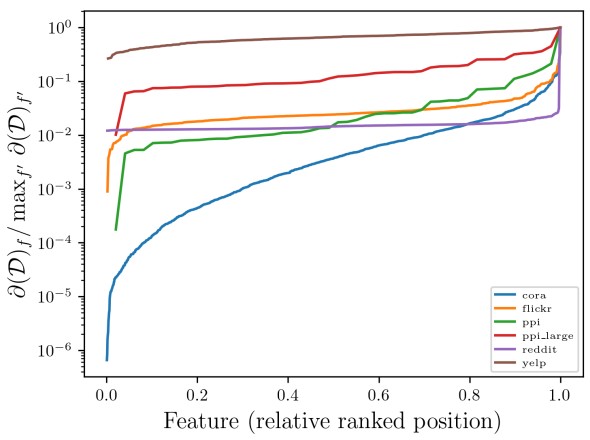

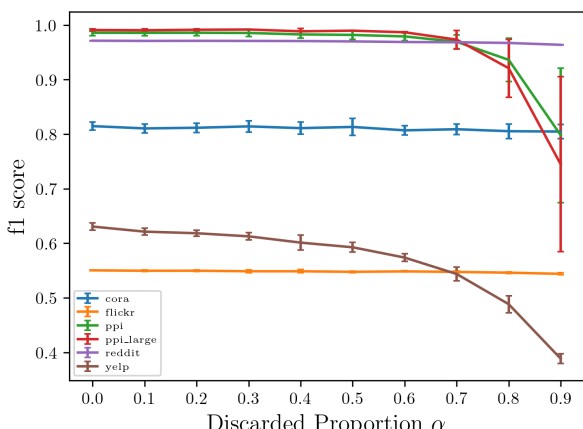

(a) Normalized intrinsic dimensionality plotted against relative ranked position of features. See Figure 3a for more detailed captions.

(b) Performance (measured by f1 score) of resulting model after discarding features from the data sets. See Figure 3b for more detailed captions.

Figure 4: Influence of Intrinsic Dimension measured through feature selection for the SAGN+SLE results.

indicates that a small number of features is responsible for almost all of the ID value. Based on these deductions, we propose the following explanation. In a certain sense, features with a low NID can be used by machine learning methods to distinguish more data points. These features may allow a learning procedure to have a more stable convergence, a shorter runtime, and a higher final performance. In our experimental study, we focus on the interplay between the ID and the achieved model performance. At this point we may note that all presented experiments are based on the same principal optimization task of stochastic gradient decent (SGD).

The observations concerning the shape of distribution of NID described above may permit to draw a connection to the simplicity bias in neural networks (Arpit et al., 2017; Shah et al., 2020; Valle-Pérez et al., 2019), the tendency of SGD to find simple models. Although further research is needed to confirm this hypothesis, the proposed association would be consistent with the assumption that SGD weights features that carry more information higher.

In our random experiment, we discard uniformly from the set of features. In every step one may lose low and high dimensional features following the distributions as shown in Figures 3a and 4a. This means for a certain discarding proportion $\alpha$ there are almost no features with low NID available. Until those disappear, SGD has the possibility to use them for obtaining the objective. But when those "good" features are not included anymore the situation changes and the performance deteriorates rapidly. On the other hand if the features are discarded in order of decreasing NID then the inevitable deterioration of model performance can be postponed for quite a bit. Conversely, when discarding features in a reverse order, e.g. ascending NID, the performance drops rapidly as the model has no access to those "good" features from the beginning.

However, we can not exclude the effect of artifacts of the method. Especially in the reverse case, where, for example, non-convergent behavior at the beginning of training triggers an early stopping condition that leads to the abortion of the optimization routine. As we regard the methods as black boxes, we have not investigated these possibilities further.

Turning to Figure 4a, we observe a few data sets that have a similar distribution of NID as those in Figure 3a. However, it seems that for some the contribution to the total NID are distributed more evenly among the features. This is particularly evident for the *yelp* data set. On the other hand, the distribution for the *reddit* data set is much more extreme, where only a small set of features have extraordinary high contribution to overall NID. This figure also clearly shows the influence of preprocessing on the NID distribution. Whereas previously in Figure 3a the line for the *cora* data set was clearly a step function, it has now become a much smoother slope. It seems that this has almost no influence on the achieved model performance in both cases.

The definitions of ID and NID indicate that these functions should have higher values for a data set than for any of its subsets, provided that the set of feature functions remains constant. This is clearly demonstrated in the graphs for both versions of the *ppi* data set, where one is the super-set of the other. It is also noteworthy that the final model performances for these data sets exhibit a high standard deviation, which is much higher than in any other experiments. It was not possible to determine the definite cause, but it seems reasonable to assume that some form of artifact of the method or mode collapse produced these high variations. The detailed figures for the remaining experiments can be found in the appendix.

We will now discuss the insights that can be gained from an overarching analysis. To do this, we will use a cross-method comparison, as only a few data sets have been processed by several papers. The aim is to identify similar distributions of NID and compare the effects on model performance. If both methods behave differently on these (possibly different) data sets, this could indicate that the NID has an effect on the methods.

The GraphSAGE and SAGN+SLE methods both use the *reddit* data set with the same preprocessing. The former shows a slight deterioration, while the latter shows almost no change in model performance.

The SAGN+SLE method is applied on the *yelp* data set, while the GCN and SGC methods are used on the *pubmed* data set. The NID distributions are quite similar, but the performance on the *yelp* data set continuously worsens with higher $\alpha$, while the performance on the *pubmed* data set remains stable until the highest discarding proportions.

Both the GCN and SGC methods use the *citeseer* data set with the same preprocessing. The first method shows very stable performance, but the second method shows minor deterioration.

The GCN, SGC and SAGN+SLE methods all use the *cora* data set. Although the preprocessing is the same in GCN and SGC, there is a clear difference in performance.

The SAGN+SLE and GraphSAGE method both use the large *ppi* data set mentioned earlier, but in both cases the performance deteriorates significantly, starting at different discarding proportions and speeds.

A possible alternative hypothesis for the above observations is that the machine learning tasks at hand can be solved using only the information provided by the adjacency matrix of the graph data. In this case, the reduction of available node features would not affect the final performance. In contrast, observations from experiments with inverse feature discarding suggest that the previous statement may not be correct, as these observations showed a much larger degradation in model performance than the other way around.

In general, this highlights a limitation of the current approach, as the chosen feature functions do not take graph edges into account. This is indirectly related to the earlier discussion on the modelling choice of what to use as the underlying set for the geometric data set. We decided to use the node features as the base set $X$ and ignored the edge features or the adjacency matrix. By using a different modelling approach, the set of feature functions could be extended or constructed entirely differently.

Taken together, these comparisons provide a strong indication that there is an effect of NID on model performance. Although we have only used the NID as an auxiliary tool to measure the ID, it shows that different methods are influenced by the ID of the data set itself. However, it is difficult to quantify the extent of this dependence on the concentration phenomenon given the present experiments on these very different methods.

At this point it is necessary to go into more detail about the graphs for the experiments for the DiffPool model. The original code accompanying the DiffPool publication uses an one-hot encoding of node label as node attributes instead of the available original node attributes. This is not stated in the paper, and the results do not seem to be easily reproducible when switching to the conventional node attributes. Nevertheless, we made this change to make it compatible with the other experiments and our method. As a result, the graph of the accuracy achieved starts much lower for zero and low discard fractions $\alpha$ than the originally claimed performance would suggest. This also explains why the results are not influenced by the discarding of features, as the node labels lack sufficient discriminatory power.

## 5.6 Summarizing the Analysis and Limitations

We present an overview of all experiments by combining the information about the intrinsic dimensionality and the model performance when features are discarded. To do this, we calculate the sum of the (approximate) normalized discriminability of the remaining features after discarding a fraction $\alpha$. The value thus obtained can be normalized by the total sum of (approximate) normalized discriminability of all features. We perform this calculation for different values of $\alpha$ and all data sets. We plot these values against the achieved performance measure (accuracy or f1 score) for the corresponding configuration. The results are depicted in Figure 5.

This leads to the striking observation that most models can cope with data sets reduced to about 30% of their original dimensionality without any loss of performance. Despite the large differences in the number of samples and features between the data sets, we do not observe a significant correlation between these features and the change in model performance.

Experiments with reversed feature selection (Figure 3) showed that there is an effect of the intrinsic dimension on the different learning methods. More specifically, we observed that this effect depends on the particular data set, as discussed in Subsection 5.5.

Our study is limited in various aspects, which we will discuss below. We observe in our investigation a low frequency of overlapping use of data sets, which is a consequence of the selection process and the underlying requirements for reproducibility. Consequently, it is challenging to compare different methods on the dimensionally reduced data sets. However, the data sets *citeseer*, *cora*, *pubmed* and *reddit* have non-trivial support over the considered papers. In this regard, we observe a similar pattern of behaviour as previously described (cf. Figure 5).

Furthermore our results build upon a rather small set of selected candidates. This could be tackled with allocating more time for achieving reproducibility per paper, which would allow for fixing or circumvent reproducibility barriers by searching for the right combination of technical tricks. It might also be possible to apply the individual methods from the publications to the other data sets as well.

The proposed method only indirectly measures the effects of the concentration of measure phenomenon through the perspective of the geometric intrinsic dimension. Moreover, is is not possible to provide an extensive overview of the complete ID influence. This is due to the fact that it is unclear if the ML methods can draw on more (complicated) feature functions than the considered ones in their processing of the data. In this case, the current restriction would be a hindrance to measuring the true impact of the ID on the methods. However, for the function class in question, we can rely on the guaranteed computability. Upon reflection, it may be not necessary to do so, as even within this restricted scope, an influence could be demonstrated.

Furthermore, the present work does not include a comparison with other approaches for measuring the influence of dimensions and any feature selection methods based on them. It is not clear if those methods would measure the same properties of the ML methods given their different underlying theoretical frameworks.

The experiments indicate that the distribution of dimensionality has a detrimental impact on the model performance. However, further experiments are required to gain a comprehensive understanding of this relationship. In particular, the measured ID is contingent upon the features functions employed. Consequently, by utilizing more comprehensive function classes, the intrinsic dimension can be more accurately determined.

Finally, we want to explain the rationale behind the decision not to utilise the correlation coefficient or an alternative methodology to quantify the direct correspondence between intrinsic dimension and model accuracy in order to facilitate a comparison of data sets. In order to do so, we will point out two theoretical aspects. Firstly, different data sets will exhibit disparate (absolute) diameters, even when employing the same feature functions. Secondly, the observable diameter is linear with regard to the feature functions, e.g. $\tau\text{ObservableDiameter}(\mathcal{D}) = \text{ObservableDiameter}(\tau\mathcal{D})$, where $\tau\mathcal{D} = (X, \{f_\tau : x \mapsto \tau f(x) \mid f \in F\}, \mu)$, for $\tau \in \mathbb{R}$ (see Hanika et al. (2022)). Thus, to consider their ratios is less informative without normalizing with regard to these properties. Therefore a first meaningful way to compare values is to focus on the ordinal relation of the feature attributes with regard to intrinsic dimension. Altogether, the experiments

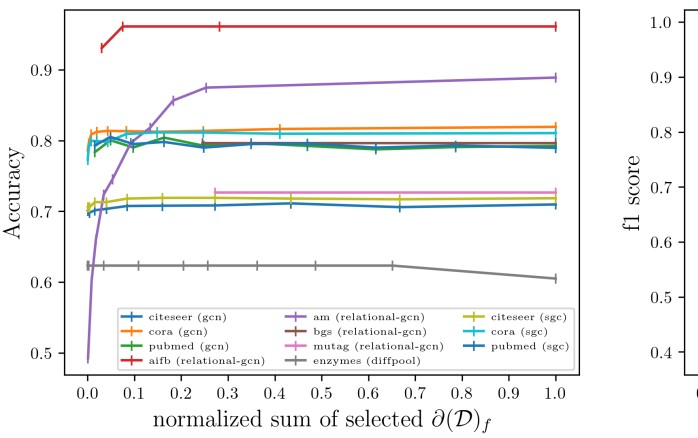

(a) Accuracy of data set and paper combinations

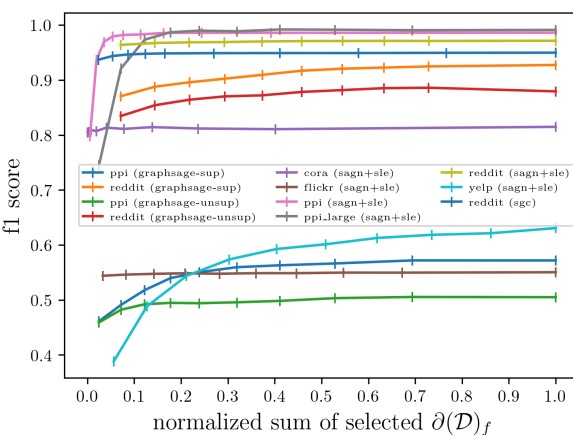

(b) f1 scores of data set and paper combinations

Figure 5: Overview of evaluation of data set and paper combinations over the remaining intrinsic dimensionality. The x-axis is the sum of the (approximated) normalized intrinsic dimensionality of the remaining features normalized by the total sum for the whole feature set. The y-axis is the resulting evaluation score obtained by the method trained on the data set with corresponding feature selection.

already show a connection between feature discarding and model performance and highlight that the order of discarding matters.

## 6 Reproducibility of the Presented Work

As a publication about reproducibility, it is only fair to also consider the own reproducibility as well. In general the reproducibility of our own work is limited by the reproducibility of the used papers. We rely exclusively on data sets provided by them and our source code is based on the one published by them. For the purposes of our experiments, it was not necessary to modify the hyperparameters of the models. In order to achieve a high degree of reproducibility, we have provided the individual scripts for the reproducibility and dimensionality experiments. These scripts include the explicitly specified environments that were used and the necessary changes to the original source code. In addition, we have endeavoured to adhere to the general guidelines set out by (Pineau et al., 2021; paperswithcode, 2021), as well as our own ontology. In total, the experiments produced approximately 600 models (6 papers, 10 factors, 10 seeds). It is not feasible to provide weights for all of them, especially if the original source code did not include sophisticated management of the paths where checkpoints were stored. Given the special form of our publication, we do not provide the model weights. The complete resources for source code, logs, and results of our experiments can be found at `https://zenodo.org/doi/10.5281/zenodo.10727907`.

## 7 Conclusions

In this work, we presented an ontology to investigate the reproducibility of machine learning research. This ontology can be used to evaluate the reproducibility of scientific publications in a standardized manner. For this, we assume that scientific evidence can be generated via theoretical evidence (for example via theorems and proofs) or via empirical evidence provided by scientific experiments. In machine learning, reproducibility research mainly focuses on the extent to which other researchers can re-execute the experiments with the same results. Consequently, our work was primarily concerned with empirical evidence, for which we have proposed a set of attributes for evaluating the level of reproducibility of a specific work. These attributes can be divided into three categories: *data set*, *software* and *computational result*.

Once reproducibility has been established, the next step is to identify the relevant influential factors that affect the outcome of the experiments. One such factor that has been the subject of the presented inves-

tigation is the intrinsic dimension, which measures the influence of the *curse of dimensionality* on a given data set. In the second part of our work, we investigated to what extent the results of empirical experiments depend on the intrinsic dimension of the data sets used for training. To provide further detail, we examined whether alterations to the intrinsic dimensionality of the employed data sets were associated with changes in model performance.

To illustrate the practical application of our ontology, we have provided detailed descriptions of how the attributes of our ontology can be used to evaluate the reproducibility of research on graph neural networks. Furthermore, we have investigated which of the well-known methods are not affected by modifications of the intrinsic dimensionality of the data sets on which they are trained on.

Our investigation indicates that attributes of reproducibility, which are part of the *data set* category of our ontology, are similarly addressed in the majority of work on graph neural networks. However, we did identify notable differences between the various methods with regard to the extend to which the data sets are documented within the README file. The quality and quantity of the documentation emerged as a crucial factors in successful reproduction with a reasonable effort. The documentation of a specific work is encompassed by our ontology within the category *software*. In general, the *software* category allowed for an accurate differentiation of the different papers with regard to their reproducibility. Attributes in the category *computational result* can be divided into two distinct groups. The first group comprises weakly separating attributes, which either represent basic reproducibility rules that are fulfilled by all of the investigated methods or strict requirements that none of the methods satisfy. The second group encompasses attributes that enable the distinction of the various methods, as these attributes are only covered by a subset of the investigated methods.

The findings presented in the second part of our study provide compelling evidence that the ID has an influence on ML algorithms. To be more detailed, our experiments show that dropping features with high individual ID has a varying impact on model performance. For example, for the GCN method, dropping high-dimensional features does not fundamentally decrease accuracies. In contrast, dropping these features when learning with GraphSAGE leads to stronger deterioration of performance. Conversely, starting with the removal of features with low individual ID results in more pronounced performance declines across all methods. This suggests that the graph learning approaches currently in use are vulnerable to changes in intrinsic dimensionality, while their resilience to the elimination of non-discriminative features (i.e., features with high individual ID) varies considerably. It has been demonstrated that the ID can be employed as a means of measuring this robustness. However, further research is required in order to ascertain the transferability of this insight. It is possible that the ID may prove to be a candidate for another meaningful attribute within the presented ontology of reproducibility.

## 8 Recommendations

Based on our findings, we recommend the following points as best practice for reproducible machine learning research.

- **Write one main script that does everything necessary.** That includes setup and or tear-down or preparation of the compute environment, downloading and preprocessing of data sets, and running of all reported experiments. It might be beneficial to chose an existing software package when handling larger data or when working on compute clusters.

- **Log all relevant information into files.** This encompasses all outcomes, utilised input variables and also selected hyperparameters, in the event that a form of hyperparameter optimisation was conducted. Often, such information is only printed to the output terminal, which hinders reproducibility.

- **Store checkpoints of all obtained models.** As multiple runs with only different seeds are required, it is imperative that the wrapping script/software ensures that no computed checkpoint is overwritten in the subsequent iteration.

- **A workflow management systems is helpful for automated aggregation and evaluation of outputs and creation of visualizations.** The reproducibility of a research paper is significantly enhanced when all steps from the experiment to the final paper, including reported results, tables, and displayed figures, can be automatically created by a dedicated script that is part of the published code. The full procedure can easily be implemented via a workflow management system.

Furthermore, we give three recommendations on how to account for the concept of intrinsic dimensionality in the scope of machine learning research.

- **Consider the intrinsic dimensionality of the used training data sets when comparing machine learning algorithms.** Our experiments indicate that well-established graph neural network approaches are significantly impaired by the increasing intrinsic dimensionality of the input data. Consequently, when comparing their performance, it is of paramount importance to ascertain the ID of the used data in order to accurately assess the extent to which performance differences are attributable to the *curse of dimensionality*.

- **Investigate, if discarding of high-dimensional features is possible for the chosen GNN.** It is possible to discard a significant fraction of features with high normalized intrinsic dimensionality from certain graph neural network models without a fundamental drop in performance. For example, when considering the GCN model, it is possible to drop up to 70% of the total number of features without decrease in accuracy, while this is not possible for GraphSAGE.

- **Account for transformations of the NID-distributions induced via preprocessing.** The preprocessing of features typically incorporates global interactions between them (i.e., averages), which alters the NID distribution in a non-trivial manner. One such case was observed in the *reddit* data set, where both SAGN+SLE and SGC applied distinct preprocessing procedures, resulting in markedly disparate NID distributions. As previously stated, this alteration of the NID distributions can profoundly impact model performance.

## 9 Limitations and Future Work

The current study is subject to certain limitations, which will be addressed in future research. Firstly, as discussed in Subsection 4.3, our ontology is limited to a fixed granularity. It has become evident that the varying depth of analysis required for reproducibility cannot be represented by the proposed ontology. Consequently, we will develop a hierarchy of ontologies with different levels of granularity by further splitting or aggregating the current attributes. For instance, in the event that a given paper presents a multitude of distinct experiments, it may be prudent to redefine attributes R2,R3 and R4 in order to encompass a broader spectrum of error ranges and fractions of non-reproducible outcomes.

Secondly, the current study is focused on a specific concept of intrinsic dimensionality. However, as discussed in Section 2, a variety of different ID estimators exist. Consequently, future work will investigate whether the reported observations are applicable to different concepts for intrinsic dimensionality.

Thirdly, our approach considers only intrinsic dimensionality for a specific feature set, namely the usual coordinate projections. However, different machine learning methods may incorporate different aspects of the data. Consequently, future work will investigate how these different aspects can be formalized as feature functions. This will lead to an ID not on for models instead of data sets. Here, the crucial problem will be the identification of a finite and computational feasible feature set which captures the model behavior.

## Acknowledgment

The authors thank the State of Hesse, Germany for funding this work as part of the LOEWE Exploration project "Dimension Curse Detector" under grant LOEWE/5/A002/519/06.00.003(0007)/E19.

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

## A    Considered Publications

In our survey, we began by collecting a list of publications to consider (Subsection 4.1). We utilized the *Semantic Scholar API*, employing the search term "graph neural network" to obtain a set of results. These results were then processed with a dedicated script to calculate the scoring metric $\frac{\text{number of citations}}{2023 - \text{year of publication}}$ for each publication. Out of the vast array of publications, we selected the top 100 papers based on these scores.

The process of reproducing the list proved to be a challenge, because of a significant degree of variability due to the non-deterministic nature of the Semantic Scholar API. This variability was particularly noticeable with regard to page ordering and the contents of the first 10000 entries. Over time we started reproducibility attempts of publications that are now no longer part of the list.

In our filtering process, we manually excluded entries that were surveys, coding frameworks, or lacked a clear connection to graph neural networks. We also ignored works that only applied Graph Convolutional Network (GCN) methods to other field of science or used time-dependent or spatial data, especially in the field of chemistry. The reason for this was that the feature selection approach used later was not directly applicable to such work, or at best uninformative. Additionally, publications applying methods to very specific data sets were not included in our list.

Some well-known papers, such as *GraphSAINT*, were not included because they did not appear under the search term used. This absence also explains why *SAGN+SLE*, *SGC*, and *GraphSAGE* do not appear in the list. To better cover the field of graph neural network research, additional search terms like "graph convolutional network" would be necessary. The subsequent changes in criteria led to a high rate of skipped publications in the full list, which was generated end of May 2023.

Table 6: All considered publications with indication on survey status. The indicators are as follows:
**i** - included in survey, **e** - excluded because experiment is not available,
**d** - excluded because method is only applied on unusual or specially build graph data sets,
**s** - skipped because of time constraints.

| Publication | Score | Status |
|---|---|---|
| Semi-Supervised Classification with GCNs (Kipf & Welling, 2016) | 2832.57 | i |
| Graph Attention Networks (Velickovic et al., 2017) | 1869.83 | e |
| How Powerful are GNNs? (Xu et al., 2018) | 913.2 | s |
| Modeling Relational Data with GCNs (Schlichtkrull et al., 2018) | 540.0 | i |
| LightGCN: Simplifying and Powering Graph Convolution Network for Recommendation (He et al., 2020) | 531.0 | s |
| Neural Graph Collaborative Filtering (Wang et al., 2019b) | 417.0 | d |
| Heterogeneous Graph Attention Network (Wang et al., 2019c) | 355.5 | s |
| Hierarchical Graph Representation Learning with Differentiable Pooling (Ying et al., 2018) | 322.4 | i |
| Graph Contrastive Learning with Augmentations (You et al., 2020) | 303.67 | s |
| KGAT: Knowledge Graph Attention Network for Recommendation (Wang et al., 2019a) | 281.5 | s |
| GCNs for Text Classification (Yao et al., 2018) | 261.4 | d |
| Link Prediction Based on GNNs (Zhang & Chen, 2018) | 248.4 | s |
| GCNs for Hyperspectral Image Classification (Hong et al., 2020) | 234.0 | d |
| DeepGCNs: Can GCNs Go As Deep As CNNs? (Li et al., 2019) | 230.75 | s |
| E(n) Equivariant GNNs (Satorras et al., 2021) | 220.5 | s |
| Weisfeiler and Leman Go Neural: Higher-order GNNs (Morris et al., 2018) | 211.0 | s |
| Predict then Propagate: GNNs meet Personalized PageRank (Klicpera et al., 2018) | 210.6 | s |
| Heterogeneous Graph Transformer (Hu et al., 2020b) | 209.0 | s |
| Heterogeneous GNN (Zhang et al., 2019) | 203.0 | s |
| Geom-GCN: Geometric GCNs (Pei et al., 2020) | 196.67 | s |
| Session-based Recommendation with GNNs (Wu et al., 2018b) | 193.0 | d |
| Self-Attention Graph Pooling (Lee et al., 2019) | 180.5 | s |
| GCC: Graph Contrastive Coding for GNN Pre-Training (Qiu et al., 2020) | 178.0 | s |
| Few-Shot Learning with GNNs (Satorras & Bruna, 2017) | 169.33 | s |
| Dynamic Edge-Conditioned Filters in CNNs on Graphs (Simonovsky & Komodakis, 2017) | 168.67 | e |

| | | |
|---|---|---|
| How to Find Your Friendly Neighborhood: Graph Attention Design with Self-Supervision (Kim & Oh, 2022) | 159.0 | s |
| Beyond Homophily in GNNs: Current Limitations and Effective Designs (Zhu et al., 2020) | 158.33 | s |
| GNN-Based Anomaly Detection in Multivariate Time Series (Deng & Hooi, 2021) | 150.0 | d |
| DKN: Deep Knowledge-Aware Network for News Recommendation (Wang et al., 2018) | 149.8 | d |
| MixHop: Higher-Order Graph Convolutional Architectures via Sparsified Neighborhood Mixing (Abu-El-Haija et al., 2019) | 142.0 | s |
| MAGNN: Metapath Aggregated GNN for Heterogeneous Graph Embedding (Fu et al., 2020) | 137.67 | s |
| Principal Neighbourhood Aggregation for Graph Nets (Corso et al., 2020) | 132.67 | s |
| Hypergraph Neural Networks (Feng et al., 2018) | 128.6 | s |
| Graph Transformer Networks (Yun et al., 2019) | 128.25 | s |
| Beyond Low-frequency Information in GCNs (Bo et al., 2021) | 125.0 | s |
| Encoding Sentences with GCNs for Semantic Role Labeling (Marcheggiani & Titov, 2017) | 123.17 | d |
| Neural Motifs: Scene Graph Parsing with Global Context (Zellers et al., 2017) | 122.67 | d |
| Graph Convolution over Pruned Dependency Trees Improves Relation Extraction (Zhang et al., 2018) | 122.4 | d |
| Graph Structure Learning for Robust GNNs (Jin et al., 2020) | 121.33 | s |
| Towards Deeper GNNs (Liu et al., 2020b) | 118.67 | s |

## B Reproducibility Context

The data collected during the successful reproducibility attempts in Section 4 can be summarized in a *Formal Context* (Ganter & Wille, 1997), where the papers make up the object set and the analyzed features of reproducibility the attribute set. A cross for paper $p$ and feature $f$ means that this feature was observed for the paper. Based on the definitions of the features, this indicates a negative aspect of reproducibility occurring in the paper.

Table 7: Formal Context derived from Reproducibility Survey with adjoining error ontology. The table is rotated sideways, e.g. Papers are objects and reproducibility features are attributes.

| | | | | GCN | Relational GCN | GraphSage | Diffpool | SGC | SAGN+SLE |
|---|---|---|---|---|---|---|---|---|---|
| computational result | predictions | | R6—no predictions | × | × | × | × | × | × |
| | | | R5—weak statistics | × | × | × | × | | |
| | | | R4—strong differences everywhere | | | | | | |
| | | | R3—strong differences in parts | | × | | | | |
| | | | R2—small deviations | × | × | × | | | × |
| | model | | R1—no model weights | × | × | × | × | × | × |
| software | source code | experiments | S19—only general idea implemented | | | | | | |
| | | | S18—hyperparameter search not included | × | × | × | × | | × |
| | | | S17—all missing | | | | | | |
| | | | S16—one missing | | | | × | | |
| | | bugs | S15—out of memory | | | | | | × |
| | | | S14—api changes | | | | | | |
| | | | S13—fix distributed through other channels | | | | | | |
| | | | S12—issue solutions not applied | | | × | | | |
| | | | S11—never fixed | | | × | | | |
| | usage | scripts | S10—unclear which version was used | | | | | | |
| | | | S9—incomplete train/test scripts | | | × | × | | |
| | | documentation | S8—missing hyperparameters | | | × | × | | |
| | | | S7—necessary arguments not clear | | | × | × | | |
| | | | S6—not up to date | × | | | × | | |
| | environment | variables | S5—important values unclear | | | | | | |
| | | | S4—seeds not set | × | | | × | | |
| | | dependencies | S3—necessary hardware unavailable | | | | | | |
| | | | S2—specified version not available anymore | | | | | | |
| | | | S1—exact version not documented | × | × | | × | | × |
| data set | transformation | selection | D11—number of samples not documented | | | | | | |
| | | | D10—train/test splits unclear | | | | × | | |
| | | preprocessing | D9—incomplete description | × | × | × | | | × |
| | | | D8—manual steps | | | × | | | |
| | availability | download | D7—on request only | | | × | | | |
| | | | D6—license restricted | | | × | | | |
| | | | D5—privacy restricted | | | | | | |
| | | | D4—access not possible | | | | | | |
| | | | D3—no direct access | | | | | | |
| | | metadata | D2—version not specified | | | | | | |
| | | | D1—format not documented | | | | | | |

## C Presentation of other Experiments

Here we want to include figures presenting the results from the influence experiments not yet presented in detail. The diagrams are structured the same way as the ones presented in Subsection 5.4 For each experiment we first show the normalized distribution of normalized intrinsic dimensionality for the used data sets (after preprocessing). For data sets with more than $10^5$ samples, an algorithm for calculating the approximated NID is used. Additionally a second figure presents the accuracy/f1 scores obtained when training on the feature reduced data sets. For a more detailed explanation see description accompanying Figure 3.

Some data sets have so few features that the steps of the discarding process are smaller than one feature. This leads to fewer data points, which in turn give the impression of only partially complete graphs for visualizations of normalized distribution of the NID or accuracy for given discarding proportions as the necessary granularity can not be achieved (see Figure 6).

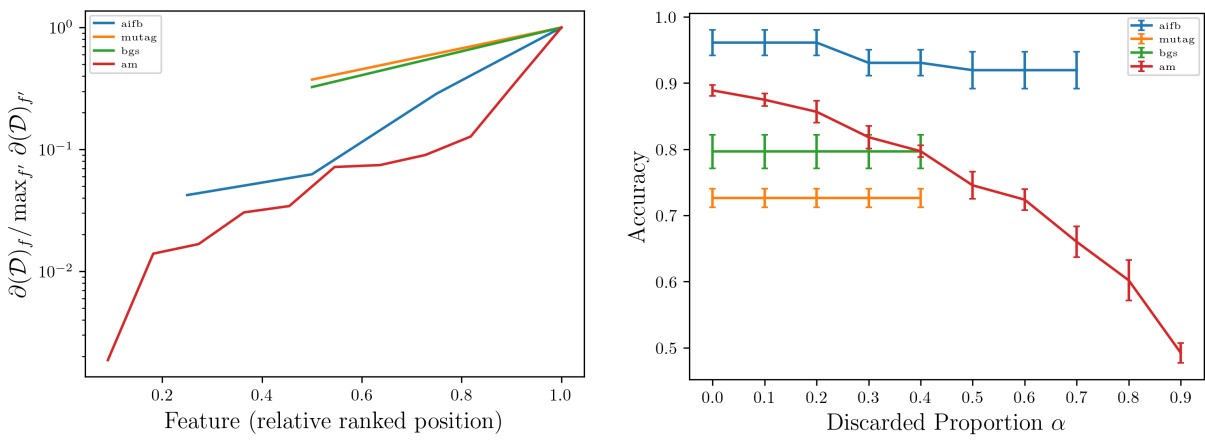

(a) Normalized intrinsic dimensionality.          (b) Performance of resulting model.

Figure 6: Experiment based on R-GCN.

The DiffPool publication used only two data sets, of which only one, namely the *enzymes* data set, has features for the graph nodes. Therefore we were not able to apply the described method to the other data set. Furthermore we encountered another problem during the DiffPool experiments (see Figure 7). It is strongly implied in the paper that the method uses the node features in its computations. However, a close examination of the source code reveals that in the default configuration, the node features are built from the classification targets of the associated graph. By changing the corresponding parameter in the training script to a different argument, which we decided on the basis of which preprocessing modifications it induced, no improvements could be observed. On the contrary, the overall performance of the method got worse. Nevertheless, we present the results obtained, which again show, that the DiffPool method does not use the node features in a comprehensible way.

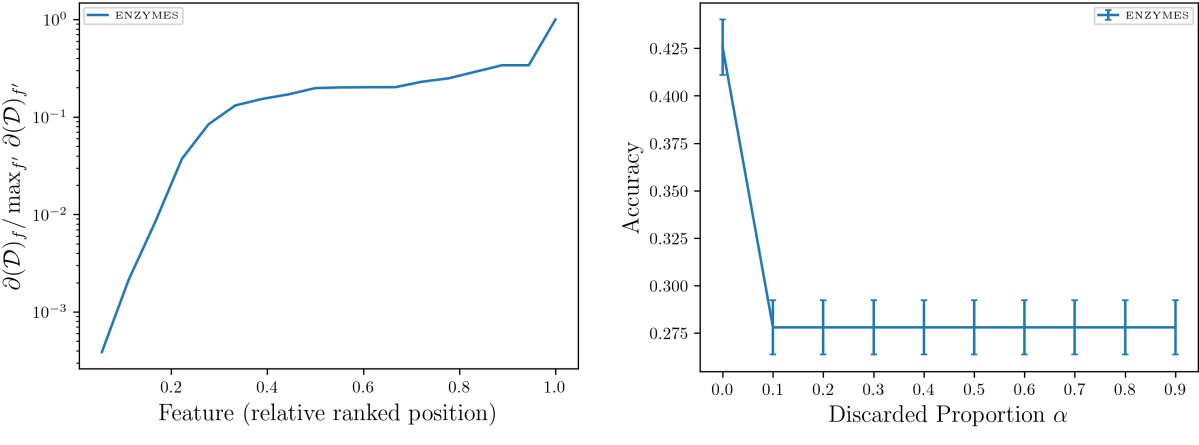

(a) Normalized intrinsic dimensionality.

(b) Performance of resulting model.

Figure 7: Experiment based on DiffPool.

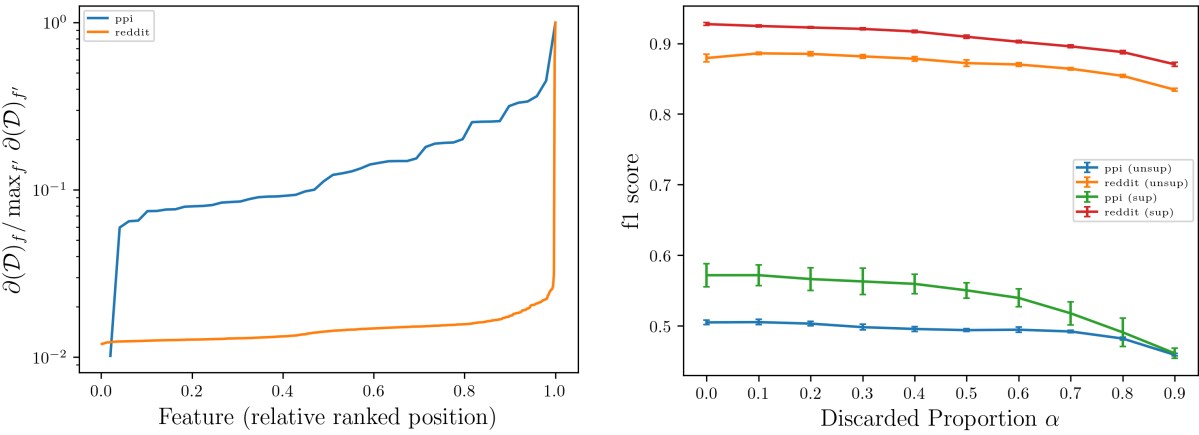

(a) (Approximated) normalized intrinsic dimensionality.

(b) Performance of resulting model.

Figure 8: Experiment based on GraphSAGE.

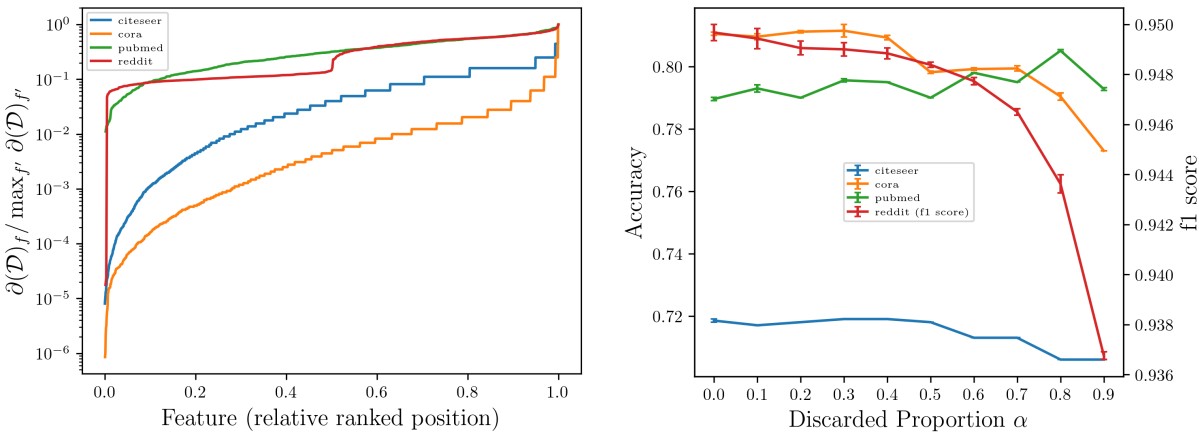

(a) (Approximated) normalized intrinsic dimensionality.

(b) Performance of resulting model.

Figure 9: Experiment based on SGC.

