# OpenReview forum: "Reproducibility and Geometric Intrinsic Dimensionality: An Investigation on Graph Neural Network Research."
_TMLR — Accepted by TMLR_

### Review · Reviewer_gYjg · 2024-04-05

**Summary Of Contributions:**

The study explores reproducibility challenges in machine learning, particularly in graph neural networks, and investigates the impact of intrinsic dimensionality on model training processes. By analyzing the methodology, results, and implications of the research, this review aims to highlight the paper's contributions and areas for improvement.

**Audience:**

Yes

**Claims And Evidence:**

Yes

**Requested Changes:**

1). Better writing to make the paper easy to follow. Possibly, tempirical validation or case studies would be helpful.

2). more discussion of representative GNN models.

**Strengths And Weaknesses:**

Strengths:
1: The paper effectively contextualizes the research within the broader landscape of reproducibility and dimensionality reduction in machine learning, demonstrating a strong theoretical foundation for the study.

2: The systematic approach to studying reproducibility in graph neural network research, including the selection of papers and the development of a scoring metric, showcases methodological rigor and attention to detail.

3: The investigation into the influence of intrinsic dimension on ML training processes and the manipulation of intrinsic dimension in graph data sets provides valuable insights for researchers and practitioners in the field.

4: By incorporating V. Pestov's geometric approach for estimating intrinsic dimension, the paper offers a novel perspective on addressing reproducibility challenges in high-dimensional ML settings.

Weaknesses:
1: While the paper presents a theoretical framework for studying reproducibility and intrinsic dimensionality, the absence of empirical validation or case studies limits the practical applicability of the findings.

2: The paper briefly touches on the challenges of implementing exact experimental procedures but lacks a detailed discussion on specific implementation hurdles faced by researchers in reproducing graph neural network results.

3: It would be better if the authors can include discussions of more representative GNN approaches, such as the following ones:
HOW POWERFUL ARE GRAPH NEURAL NETWORKS? ICLR'19
Learning to drop: Robust graph neural network via topological denoising. WSDM'20.
Robust Graph Representation Learning via Neural Sparsification. ICML'19

4: Some sections of the paper could benefit from improved clarity and organization to enhance readability and facilitate understanding for a broader audience. Basically, the current writing is hard to follow.

---

> ### Author Response · Authors · 2024-06-09
> **Response**
>
> Dear Reviewer gYjg,
>
> Thank you for taking the time to review and comment upon our submission. We
> found the advice very constructive and have adapted our manuscript with respect
> to your suggestions.
>
> We respond to each comment individually below.
>
> - 1: We agree, that such a study with regard to the reproducibility ontology
>   would be more insightful. Unfortunately, this would require to develop a
>   scientific evaluation framework or implementing a survey of experts. We can
>   already say that several elements of our ontology are based on existing
>   approaches. Hence, we already have a partial validation, albeit a one that can
>   be enhanced. As suggested by reviewer *1k1o*, the ontology could be useful for
>   a broader machine learning context. Therefore, the survey of experts, ideally
>   in a later, more comprehensive investigation, would be a good idea.
>   Unfortunately, this is not feasible within the scope of the work presented.
>
> - 2: We extended the discussion on specific implementation hurdles.
>
> - 3: We would have liked to include the suggested papers, but we have not found
>   any original implementations, except for ``How powerful are graph neural
>   networks?''. For this paper, the reproducibility does not seem to be very good
>   according to the issues in the repository. Even the author refers several
>   times to another re-implementation in a library. In our work, however, we
>   wanted to stay as close as possible to the original scientific
>   implementations.
>
> - 4: We have rewritten several paragraphs.
>
> According to the TMLR guidelines we will upload the revised manuscript after
> receiving the third review.
>
> Thank you again for your thoughtful comments.
>
> Sincerely,\
> The Authors

---

### Review · Reviewer_1k1o · 2024-04-21

**Summary Of Contributions:**

In this article, the authors propose a new ontology, taking the form of a set of guidelines organized into questions, for assessing and evaluating the reproducibility of machine learning projects. This ontology is very general (i.e., not specific to particular fields of data science), and covers an exhaustive list of aspects related to experimental claims and evidence, including (but not limited to) availability of source code, data sets and documentation. The authors then applied their ontology to graph neural networks, and, among the papers that were deemed reproducible enough, investigated the influence on the intrinsic dimension of data sets on the different model performances. By characterizing how data set features concentrate (i.e., aggregate towards their mean or median), and removing them progressively, they were able to detect those models that are very sensitive to these removals (e.g., GraphSAGE), and those that are more robust (e.g., Graph Convolutional Networks).

**Audience:**

Yes

**Broader Impact Concerns:**

No concerns.

**Claims And Evidence:**

Yes

**Requested Changes:**

---There a few sentences/paragraphs that I could not understand. Could the authors rephrase or explain in greater details?
p11: "Furthermore, the collection [...]"
p14: "Such a function allows for estimating [...]"
p17: "For this, we first extract the logic [...]"

---p9, criterion R5: setting a threshold at 5 samples sounds arbitrary. Is there some justification? Also, I think this is not very relevant for some machine learning fields such as one-shot learning, where learning data is very restricted.

---p15: I do not think it is worth introducing Borel measures in Definition 1, as they are not used later.

---p15: d_F is only a pseudo-metric, not a proper metric.

---Here is a (non exhaustive) list of typos:
p1: "isto"
p3: "with in influence"
p3: "those those"
p5: "now" --> new
p7: "sideeffects"
p7: "me" --> be
p8: "outline" --> outlined
p10: "We starting"

**Strengths And Weaknesses:**

A clear strength of this work is the rigorous ontology (and corresponding pipeline) that has been proposed. It is complete and exhaustive, and, I believe, very useful for all machine learning practitioners who care about reproducibility. Its application to graph neural network projects is also quite interesting, and allows a clear high level picture of the different approaches and how easy it is to re-use (some of) them. Hence, I think this article provides a good contribution to the machine learning field.

I do have a few concerns though:

---The writing could be greatly improved. I had a lot of trouble understanding some paragraphs (see also below), and there are numerous confusing typos in the text.

---The investigation of intrinsic dimension is not so compelling. I think that the intrinsic dimension has been investigated under various other lenses (for instance, see https://proceedings.neurips.cc/paper/2020/file/54f3bc04830d762a3b56a789b6ff62df-Paper.pdf), and that a proper comparison (or at least some discussion in the text) should be implemented in order to better highlight the specificity of the dimension used in this work. I also find it a bit difficult to grasp the message conveyed by this last section: how should I interpret that GCN is more robust than GraphSAGE for instance? Should I understand that I can safely discard the high-NID features when using GCN models (and thus save on running time and memory), while it is dangerous for GraphSAGE? This should appear more explicitly.

---Related to this comment, I think the proposed ontology could be improved by also designing, e.g., some sort of reproducibility score, in order to formally compare and rank papers. Otherwise, it is a bit unclear how the question tree derived from the ontology can help in trimming the papers from the literature.

---

> ### Author Response · Authors · 2024-06-09
> **Response**
>
> Dear Reviewer 1k1o,
>
> Thank you for taking the time to review and comment upon our submission. We
> found the advice constructive and have incorporated many of the suggestions into
> our revision.
>
> As well as rewriting the suggested paragraphs and correcting typos, we respond
> to the comments individually below:
>
> - Criterion R5 was poorly worded and therefore ambiguous. We thank the reviewer
>   for catching this. We require that for a given configuration, the experiment
>   should be run (at least) five times so that the average of the results gives
>   a higher confidence. We adapted the text accordingly.
>
> - We thank the reviewer for pointing out the inconsistency in Definition 1. Of
>   course, $d_F$ is only pseudo-metric for general choice of $F$. We have
>   corrected this error and adapted Definition 1 to our specific finite case.
>
> - We were not aware of the proposed paper. Its content is about deep
>   convolutional networks and how they deform the sample densities of the data.
>   The work refers to the intrinsic dimension of activations in different layers.
>   The connection to our work would require a much broader formal framework, as
>   we have only investigated the intrinsic dimensionality of the data sets with
>   respect to the learning features of a model. However, our investigation does
>   not target the learning process or the learned model itself. We have therefore
>   decided not to make an experimental comparison, but we have revised the
>   relevant paragraphs and hope that it is now even clearer what the specifics of
>   the intrinsic dimension used are, and that the point has been answered.
>
> - You have understood the robustness statement correctly. To make this more
>   explicit, we adapted the text accordingly.
>
> - We have kept the evaluation through the ontology qualitative so far. This
>   results in an ordinal measuring structure that allows formal comparison and
>   ordering. We agree that it would be helpful to derive a score function.
>   However, at this moment we are not able to define a score valuation function
>   for which we can guarantee that it is meaningful. Moreover, the ordinal
>   evaluation framework also allows us to explain differences in reproducibility
>   or lack thereof.
>
> According to the TMLR guidelines we will upload the revised manuscript after
> receiving the third review.
>
> Thank you again for your thoughtful comments.
>
> Sincerely,\
> The Authors

---

### Review · Reviewer_qkD2 · 2024-06-24

**Summary Of Contributions:**

The contribution of the paper is twofold.

First, it presents an ontology to determine if a machine-learning paper is reproducible and applies the proposed ontology to analyze the reproducibility of papers in the field of graph convolutional networks. \
Second, it analyzes the effect of a concentration-based estimate of the intrinsic dimension (ID) on the model performance. In this second part, the authors mainly focus on the impact of an ID-based feature selection on model accuracy.

**Audience:**

Yes

**Broader Impact Concerns:**

No ethical implications or concerns would require adding a broader impact for the current work.

**Claims And Evidence:**

No

**Requested Changes:**

1. A convincing answer to questions Q1 and Q2 in the weaknesses section is critical for acceptance.

2. In particular, a quantitative connection between the dataset's intrinsic dimension (not just normalized by the maximum dataset value) and reproducibility, or at least its impact on model performance, is critical for acceptance.

3. How do the results shown in section 5.3 depend on the specific choice of feature functions? The authors in the paper discuss the case of $f_j(x) = x_j$. It would be nice to show that the results do not substantially depend on this choice.

**Strengths And Weaknesses:**

**Strengths**

The paper proposes a collection of requirements to judge whether an ML paper is reproducible. This contribution is a valuable attempt for the community, which still lacks a shared set of guidelines to evaluate the reproducibility of ML studies.


**Weaknesses**

The first part of the paper, which concerns reproducibility, and the second, which focuses on feature selection, have very little in common to the point that they could be regarded as two standalone papers.  \
When the data ID is high, a ML algorithm might generalize poorly out-of-sample, i.e., the test error might be higher, but a high test error does not make a ML study less reproducible. Indeed, the value of the test error can be perfectly reproduced.

> Q1: *What is the connection between a machine-learning paper's reproducibility and the data's intrinsic dimension?  Are, for instance, results obtained from high dimensional data systematically less/more reproducible than those obtained from low dimensional data?*

The authors claim that Section 5's main goal is to “investigate the influence of intrinsic dimensionality on model behavior.” However, Section 5 discusses the impact of feature selection on model performance. The ID is never directly linked to the model performance, and its value is never shown: in Figs 3a and 4a, the ID is normalized so that its value always goes from 0 to 1.

> Q2: *Is there any evidence in the experiments shown in the paper that the data intrinsic dimension correlates with model performance?*

On a high level, the section shows that feature selection can be applied without decreasing accuracy. For example, if one keeps 40% of the features (Fig 3/4b and Fig 5), the accuracy is not compromised. This is a well-known fact (see, e.g., https://link.springer.com/article/10.1007/s10462-019-09682-y).

> Q3 *What kind of new knowledge or perspective does the proposed ID-based feature selection bring us?*

---

> ### Author Response · Authors · 2024-07-04
> **Response**
>
> Dear Reviewer qkD2,
>
> Thank you for taking the time to review and comment upon our submission. We
> found the advice very constructive and have adapted our manuscript with respect
> to your suggestions.
>
> The aim of this project is to unveil hidden effects that influence model
> performance. To investigate these scientifically, we created the reproducibility
> framework to control for the influence of other sources. This approach could be
> perceived as similar to the method of differential diagnosis. We adapted the
> manuscript to emphasize this more clearly.
>
> On the other hand, the ID itself can be understood as an aspect of
> reproducibility in a broader sense, e.g., are results reproducible for different
> data sets from the same domain? Particularly it is an important estimate for the
> extent to which an approach is transferable to another data set. For example, it
> could be the case that a performance measure is reproducible for small ID
> datasets, but not for large ID datasets. In the medium term, it is conceivable
> that it could become an independent attribute/dimension of reproducibility.
> However, to derive this is deemed future work.
>
> Our work provides new insights through empirical results related to the above
> points using feature selection. As we indicated in the discussion, another
> important perspective could be a new way to measure the robustness of a machine
> learning model.
>
> The extent to which the intrinsic dimension correlates with model performance
> was not investigated. Instead, evaluations that capture a similar notion were
> included. For each value of the discarding factor in Figures 3b/4b, the
> corresponding value of the intrinsic dimension in Figures 3a/4a can be found.
> However, a figure showing this mapping directly was not included. The primary
> reason for this is that utilising a correlation coefficient or an alternative
> methodology to quantify the direct correspondence between these two variables
> entails certain theoretical limitations.
>
> Different data sets will exhibit disparate (absolute) diameters, even when
> employing the same feature functions. Furthermore the observable diameter is
> linear with regard to the feature functions, e.g.
> $\tau\text{ObsDiam}(\mathcal{D})=\text{ObsDiam}(\tau \mathcal{D})$, where $\tau
> \mathcal{D}=(X, set(f_\tau: x\mapsto \tau f(x)\mid f\in F), \mu)$ (see
> Hanika2022). Therefore a meaningful way to compare values is to focus on the
> ordinal relation of the feature attributes with regard to intrinsic dimension.
> In contrast, to consider their ratios is less informative. The applied
> normalization in Figures 3a and 4a was merely chosen for enhancing the
> visualization. Altogether, the diagrams show a connection between feature
> discarding and model performance and highlight that the order of discarding
> matters.
>
> We were not aware of the suggested paper. The referenced thresholds are only
> mentioned in the text, whereas their Table 5 does not specify the percentage for
> which the model performance is reported. However, only the best result is
> reported. In general, the order of removal is important and leads to very
> different behavior. We have shown (Figure 3) that a very small percentage (1%)
> of features has a significant influence.
>
> Your final observation regarding the dependency of the results on the specific
> choice of feature functions is accurate. If, for example, a learning procedure
> is able to use all 1-Lipschitz functions, then the ID results will be different,
> in particular the ID will be lower. Our selection of feature functions attempts
> to capture the capabilities of the most commonly used learning methods in
> current research. However, we are open to other ideas in this direction.
>
> Thank you again for your thoughtful comments.
>
> Sincerely,\
> The Authors

---

### Decision · Action_Editor_SnDs · 2024-09-13

**Recommendation:** Accept with minor revision

**Comment:**

The paper provides a rigorous ontology to test reproducibility of papers in ML, and will thus potentially be a meaningful reading for all researchers processing large data bases. The proposed method is tested on multiple papers with success. The reviewers all agree of the interest to the TMLR community and beyond. However, there is also agreement on the fact that the paper would benefit greatly of some improvement in the writing. I this recommend acceptance, conditional on the authors taking this last comment into account, who would thus publish a final version with an improved presentation.

**Audience:**

The paper concerns an issue that all researchers in the field of machine learning (and beyond) will face: reproducibility. It is then clear, as pointed by the reviewers, that the broad audience of TMLR will potentially be interested.

**Claims And Evidence:**

The authors have used their proposed methodology for estimating reproducibility on a base case made of 6 papers. They have also linked the level of reproducibility to the data intrinsic dimension, which was also investigated on these base cases.